# Capsid protein structure in Zika virus reveals the flavivirus assembly process

Ter Yong Tan[1,2,7], Guntur Fibriansah [1,2,7], Victor A. Kostyuchenko[1,2], Thiam-Seng Ng[1,2], Xin-Xiang Lim[3], Shuijun Zhang[1,2], Xin-Ni Lim [1,2], Jiaqi Wang[1,2], Jian Shi[4], Marc C. Morais[5], Davide Corti [6] & Shee-Mei Lok [1,2✉]

Structures of flavivirus (dengue virus and Zika virus) particles are known to near-atomic resolution and show detailed structure and arrangement of their surface proteins (E and prM in immature virus or M in mature virus). By contrast, the arrangement of the capsid proteins: RNA complex, which forms the core of the particle, is poorly understood, likely due to inherent dynamics. Here, we stabilize immature Zika virus via an antibody that binds across the E and prM proteins, resulting in a subnanometer resolution structure of capsid proteins within the virus particle. Fitting of the capsid protein into densities shows the presence of a helix previously thought to be removed via proteolysis. This structure illuminates capsid protein quaternary organization, including its orientation relative to the lipid membrane and the genomic RNA, and its interactions with the transmembrane regions of the surface proteins. Results show the capsid protein plays a central role in the flavivirus assembly process.

[1] Programme in Emerging Infectious Diseases, Duke–National University of Singapore Medical School, Singapore 169857, Singapore. [2] Centre for BioImaging Sciences, Department of Biological Sciences, National University of Singapore, Singapore 117557, Singapore. [3] Department of Biological Sciences, National University of Singapore, Singapore 117558, Singapore. [4] CryoEM Unit, Department of Biological Sciences, National University of Singapore, Singapore 117557, Singapore. [5] Department of Biochemistry and Molecular Biology, Sealy Center for Structural and Molecular Biophysics, University of Texas Medical Branch, Galveston, TX 77555-0647, USA. [6] Humabs BioMed SA, a subsidiary of Vir Biotechnology, Inc., CH-6500 Bellinzona, Switzerland. [7] These authors contributed equally: Ter Yong Tan, Guntur Fibriansah. ✉email: sheemei.lok@duke-nus.edu.sg

Zika virus (ZIKV), a mosquito-borne flavivirus, usually causes asymptomatic infection; however, severe neurological complications such as Guillain-Barré syndrome and congenital Zika syndrome may arise[1]. Other major flavivirus human pathogens are dengue virus (DENV), West Nile virus (WNV) and tick-borne encephalitis virus (TBEV).

ZIKV is an enveloped positive-sense single-stranded RNA virus. During infection, the envelope (E) proteins on ZIKV bind to host cell receptors and the viral particle is then endocytosed. The E proteins facilitate fusion of the virus with the endosomal membrane, resulting in the release of the genomic RNA into the host cell cytoplasm. Translation of the RNA genome occurs in the endoplasmic reticulum (ER). The RNA is translated as a single polypeptide chain encompassing all the viral proteins: C-prM-E-NS1-NS2A-NS2B-NS3-NS4A-NS4B-NS5. The polypeptide chain is threaded back and forth through the ER membrane, exposing different viral proteins within the sequence to either the cytoplasmic or ER lumen (Fig. 1a). Host and viral proteases then cleave the polyprotein into three structural (C, precursor membrane (prM), and E) and seven non-structural proteins (NS1, NS2A, NS2B, NS3, NS4A, NS4B, and NS5). The structural proteins are used to assemble new virus shells, whereas the non-structural proteins are involved in the replication of the viral RNA genome[2,3].

Although the capsid protein is known to interact with the viral RNA genome to facilitate packaging into the virus particle, it is not clear how encapsidation occurs. The 122 amino acid ZIKV capsid protein is the first protein translated in the single viral polypeptide chain (Fig. 1a). The N-terminal end of the capsid protein contains a highly positively charged loop followed by five α-helices (helices α1–α5) (Fig. 1b). The helix α5 of the capsid protein, which immediately precedes the prM protein in the viral polyprotein, is a signal peptide that serves to direct the translocation of prM through the ER membrane into the ER lumen. Within the ER lumen, host signal peptidase cleaves at the capsid helix α5–prM junction, resulting in a membrane-bound anchored

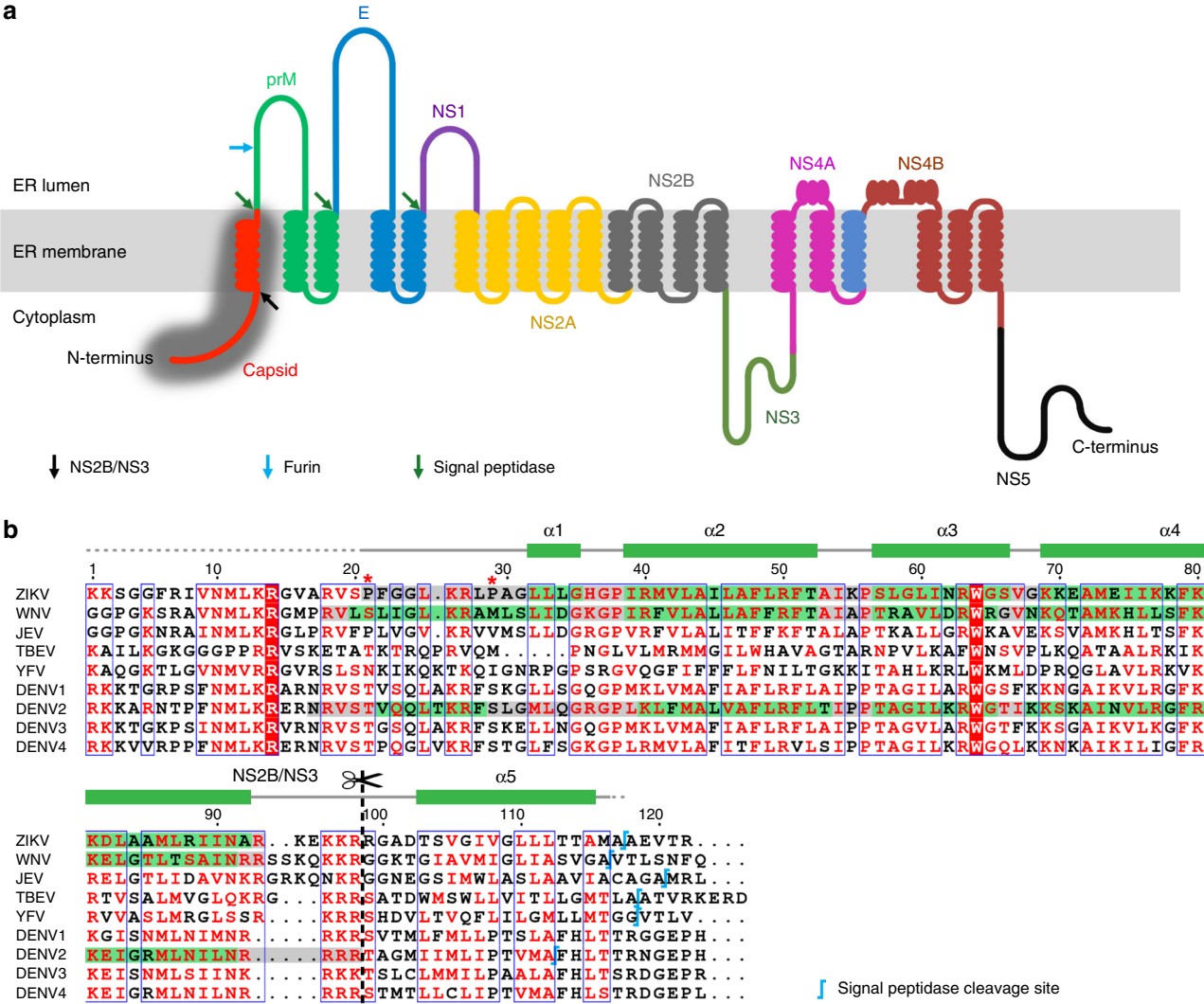

**Fig. 1 Capsid protein topology and sequence. a** Topology of the translated single ZIKV polyprotein chain. The full-length capsid protein, for which the cryoEM structure is determined here, is highlighted in dark gray. **b** Sequence alignment of capsid proteins across flaviviruses. Above the amino acid sequences, the green rectangles represent the helical structures, whereas the gray lines represent the loops in the capsid structure of our cryoEM structure of immZIKV complexed with DV62.5. Dotted lines indicate parts that could not be observed. Similar coloring codes were used to box the sequences of residues forming the secondary structures observed in the crystal structures of the capsid proteins of ZIKV (PDB ID:5YGH) and WNV (PDB ID: 1SFK), and NMR structure of the capsid protein of DENV (PDB ID: 1R6R). The NS2B/NS3 and the signal peptidase cleavage sites[49,50] are also indicated. Two proline residues that interrupt the formation of helix α1 leading to a shorter version in the ZIKV compared with the other flavivirus are indicated by red asterisks.

capsid. At the cytoplasmic face, the viral protease NS2B/NS3 cleaves at the capsid helix α4–α5 junction to release mature capsid protein (Fig. 1b), leaving behind the helix α5 embedded in the ER membrane. It is known that the cleavage of signal peptidase and NS2B/NS3 at their respective ends of helix α5 occur sequentially. Presently, it is believed that only the mature capsid protein, which consists of residues from its N terminus to the end of helix α4, are involved in flavivirus assembly. Thus, all available atomic resolution nuclear magnetic resonance (NMR) and crystal structures of isolated recombinant flavivirus capsid proteins reflect that truncated form: DENV (residues 21–100)[4], WNV (residues 24–96)[5], and ZIKV (residues 26–98)[6,7], and also determined in the absence of lipids and viral RNA. This limits our understanding of how capsid interacts with the other components within the virus particle. These structures showed capsid helices α1–α4 connected by loops. The crystal structures of ZIKV capsid protein show it has a similar tertiary organization as the WNV and DENV capsid proteins, although the orientation of helix α1 with respect to the other helices is more similar to that of the WNV capsid protein (Fig. 1b). The capsid protein exists as a homodimer with characteristically distinct surfaces on two opposing sides. One side with hydrophobic helices α1 postulated to interact with viral membrane, whereas the other side with highly positively charged helices α4 postulated to interact with viral RNA.

After translation of the viral polyprotein, the capsid proteins are on the cytosolic side of the ER (Fig. 1a), where they form a complex with viral RNA genome. This complex then buds into the ER lumen, acquiring a lipid membrane with anchored viral prM and E proteins, and thus forming the immature ZIKV (immZIKV) particle.

The icosahedral surface of the immZIKV consists of 180 copies of prM–E heterodimers (Supplementary Fig. 1a) assembled into 60 spikes (Supplementary Fig. 1b), each made from trimers of the prM–E complex[8] (Supplementary Fig. 1a). The E protein consists of three domains: E-DI, E-DII, and E-DIII, and the prM protein consists of pr and M protein. In each icosahedral asymmetric unit (asu) of the virus shell, there are three individual prM–E complexes, colored in shades of red, green, and blue in Supplementary Fig. 1b. The surface proteins are organized in a unique intertwined conformation between the spikes (Supplementary Fig. 1c). The top of each trimeric spike, formed by E-DIIs and the pr portion of prMs (Supplementary Fig. 1c), is underpinned by the base, which is formed by E-DIs, E-DIIIs, and the M moiety of prM from adjacent spikes (Supplementary Fig. 1d). Thus, the base is also a trimeric arrangement of prM–E complexes, each belonging to three different spikes, with their transmembrane (TM) segments immediately located underneath.

During the maturation process through the trans-Golgi network, the trimeric prM–E spikes on the immZIKV rearranged into prM–E dimers[9], which are lying flat on the virus surface. This structural change facilitate furin cleavage of pr from the prM. When virus is released to the extracellular environment, the pr fall off the particle resulting in the fully infectious mature virus. The near-atomic resolution cryo-electron microscopy (cryoEM) structure of the mature ZIKV particle[10,11] shows the E-protein dimers lying parallel to each other forming a raft. Thirty of these rafts are organized in a herringbone pattern on the virus surface (Supplementary Fig. 2).

Thus far, the flavivirus assembly process at the ER remains largely unknown. In cryoEM images, empty virus particles are rarely observed for both mature and immature virus preparations. This indicates that there are likely interactions between the inner core, containing the capsid–RNA complex, and the inner leaflet of the virus bilayer lipid membrane. However, all available cryoEM structures of the mature flaviviruses[10,12,13] show very poor capsid–RNA densities, even as their surface proteins are resolved to near-atomic resolution. Previous immature DENV structures also did not show obvious densities corresponding to capsid protein[14–16]. Interestingly, a recent report of an overall 9.5 Å resolution icosahedrally averaged immZIKV cryoEM map shows a diffused low-resolution blobs of densities located underneath the inner leaflet of the lipid bilayer membrane[8], whose volume may correspond to capsid proteins.

Here we stabilize capsid proteins in the immZIKV by complexing the particle with the antigen-binding fragment (Fab) from the human monoclonal antibody (HMAb) DV62.5, which binds across prM and E proteins. Antibody binding results in a significantly improved resolution of the capsid protein densities within the virus particle in a cryoEM reconstruction of the virus: Fab complex, likely resulting from reducing the inherent dynamics/flexibility of the capsid proteins. This structure reveals the quaternary organization of the capsid proteins, their relative orientation with respect to the viral lipid membrane and the RNA genome, and their interactions with the transmembrane regions of the surface prM and E proteins. These results suggest that the capsid protein has a strong influence on the organization of the overall quaternary structure of the virus particle and thus plays a central role in the virus assembly process.

## Results

**Fab DV62.5 binding improved capsid protein densities.** HMAb DV62.5 was previously shown to cross-react to all four serotypes of DENV and ZIKV, and is non-neutralizing[17]. Here we determined the icosahedrally averaged cryoEM structure of an immZIKV complexed with a Fab DV62.5 (Fig. 2a, left panels, and Table 1) by single-particle analysis (SPA). The overall resolution of the map is 8 Å, as determined by the gold standard Fourier shell correlation method (Fig. 2a, top right panel). The resolutions of prM–E layer and the capsid protein layer of the map are ~6.5 Å (Supplementary Fig. 3a) and ~9.5 Å resolution (Fig. 2a bottom right panel and Supplementary Fig. 3b), respectively. The capsid protein densities have some tubular shapes corresponding to helices present in the crystal and NMR structures of the capsid proteins (Fig. 2c, two left panels, and Supplementary Movie 1). We also determined the structure of the uncomplexed immZIKV to overall 9 Å resolution (Fig. 2b and Table 1). Consistent with the uncomplexed immZIKV structure previously solved by Prasad et al.[8], the capsid protein density for this structure is poor, indicating either inherent flexibility of capsid proteins and/or different relative positions in individual particles (Fig. 2b, bottom left panel, and Supplementary Fig. 4, bottom panels). These results indicate that Fab DV62.5 stabilizes the immZIKV capsid proteins, resulting in a well-resolved capsid density to allow fitting with the crystal structure of capsid protein (Fig. 3a and Supplementary Movie 1). However, the density of the capsid protein layer compared with that of the prM–E is slightly weaker, as the capsid layer disappeared at lower visualization thresholds than the prM–E layer. We collected tomograms of immZIKV:Fab DV62.5 complex and calculated an ~16 Å resolution subtomogram asymmetric averaged map that shows that the capsid proteins do not have full occupancy at some of the icosahedral symmetry equivalent positions in the capsid protein layer (Supplementary Fig. 5 and Supplementary Movie 2). Fourier shell correlation (FSC) calculation between each of the 60 capsid densities in the C1 subtomogram-averaged map and the density from the icosahedral-averaged SPA cryoEM map showed that most of the capsid densities from C1 subtomo-averaged map correlate to the density from the SPA cryoEM map to resolution better than 18 Å, with some capsid densities even reaching ~10 Å (Supplementary Fig. 6a, b). There are also capsid densities with

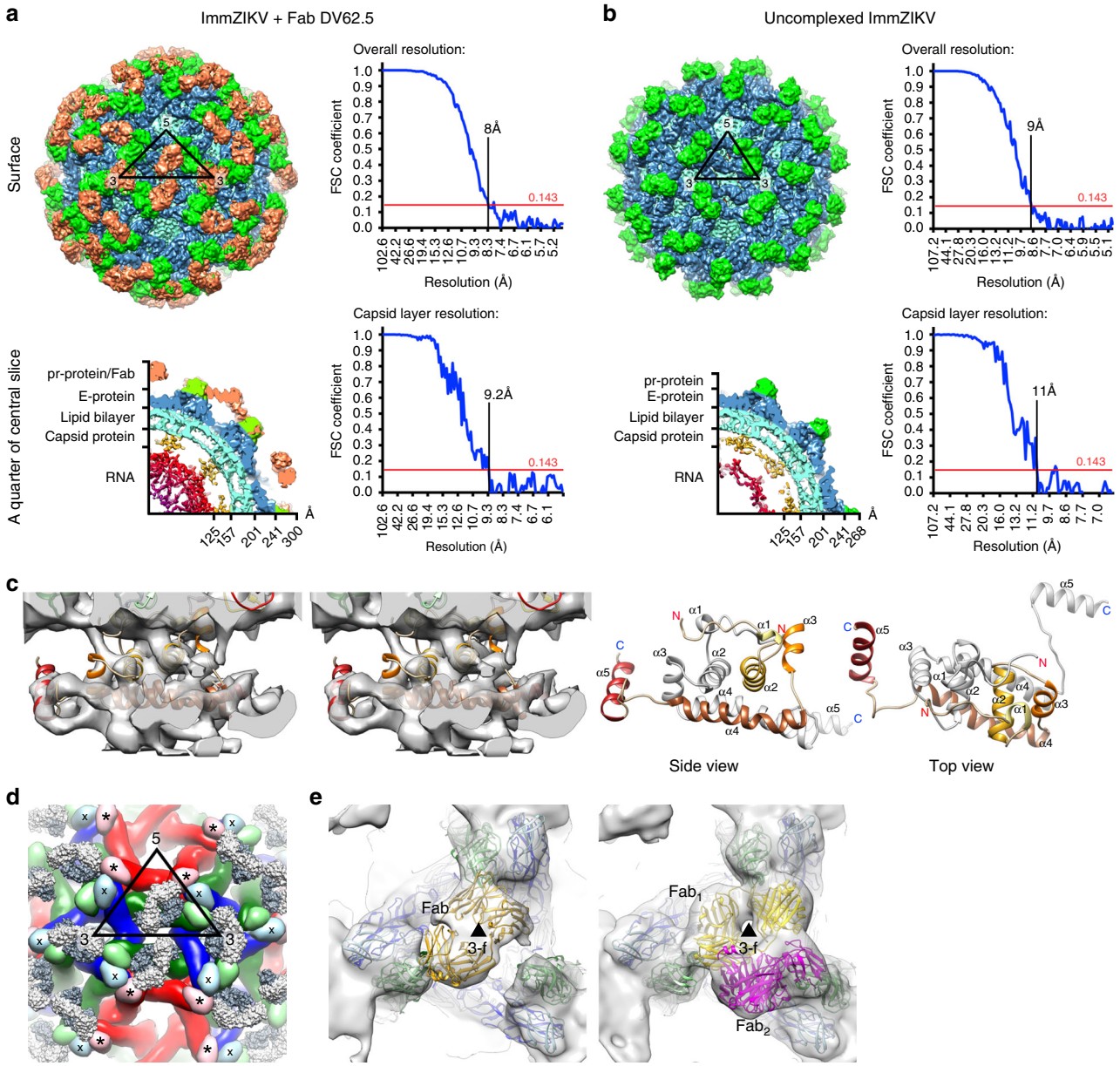

**Fig. 2 ImmZIKV structure is stabilized by DV62.5, improving resolution of the capsid protein density.** The surface (top panels) and a quarter of a central slice (bottom panels) of the cryoEM maps of **a** immZIKV:DV62.5 and **b** uncomplexed immZIKV. The central slice shows the capsid protein bridges the RNA and the inner leaflet of the lipid membrane. The overall resolution of the maps (top) and the capsid layer regions (bottom) are shown. Densities corresponding to Fabs are colored in orange. **c** Overall structure of a capsid dimer with each protomer containing helices α1 to α5 (colored from light to dark brown shades). Stereo-diagram (two left panels) of the fitted capsid dimer (ribbon) in its corresponding density (gray) at a contour level of 1.65. Different views of capsid dimer structure (two right panels) are shown. The N terminus (red N letter) and C terminus (blue C letter) of the capsid protein are indicated. One of protomer is colored in gray. **d** Occupancies of the Fab DV62.5 on the immZIKV. Left panel, on immZIKV, Fab DV62.5 (gray surface representation) binds to all red prM–E complex. The epitope on the blue prM–E is completely occluded by neighboring prM–E molecules; therefore, no Fab binding is detected. Around the threefold axis, although the epitope is completely exposed in all green prM–E-protein molecules, the space at threefold vertices is likely unable to accommodate three Fabs. E-prM proteins with bound Fab DV62.5 at full occupancy are indicated by asterisks (*), whereas those with no binding by crosses (×). The three individual E proteins in an asu, each located adjacent to either five-, two- or threefold vertices, are colored in red, green, and blue, respectively, and their binding partner, the prM molecule in a lighter shade of the same color. **e** Localized reconstruction on the densities surrounding the threefold vertices showed two major Fab binding classes. Left panel, the first class of Fab binding shows only one Fab (yellow) around this vertex. Right panel, the second class of Fab binding showed two Fabs bound (yellow and magenta). The E–prM proteins and Fab molecules are drawn in ribbon representation.

correlation > 20 Å. These low correlations are from the capsid protein dimer with partial occupancy or even missing, leading to poor densities. In the SPA cryoEM map, this partial occupancy likely explains the weaker density of the capsid protein compared with the prM–E layer. We first fitted the crystal structure of the

ZIKV capsid protein[7] into the 9.5 Å resolution capsid layer density of the Fab DV62.5:immZIKV complexed map obtained from SPA (Fig. 3a) by rigid body fitting. To optimize the fit, individual helices were allowed to move independently to a small degree, to better fit the experimentally obtained density map;

**Table 1 CryoEM data collection, refinement, and validation statistics.**

| | ImmZIKV-Fab DV62.6 complex [SPA] (EMDB-0932) (PDB 6LNT) | ImmZIKV (EMDB-0933) (PDB 6LNU) | ImmZIKV-Fab DV62.5 complex [StA] (EMDB-0933) |
|---|---|---|---|
| Data collection and processing | | | |
| Magnification | 47,000 | 59,000 | 47,000 |
| Voltage (kV) | 300 | 300 | 300 |
| Electron exposure (e–/Å$^2$) | 18 | 18 | 90 |
| Defocus range (μm) | −0.5 to −3.5 | −0.5 to −3.5 | −3 to −4 |
| Pixel size (Å) | 1.71 | 1.34 | 1.71 |
| Symmetry imposed | I | I | C1 |
| Initial particle images (no.) | 84,924 | 79,701 | 1934 |
| Final particle images (no.) | 19,295 | 7,922 | 1152 |
| Map resolution (Å) | 8 | 9 | 13 |
| FSC threshold 0.143 | | | |
| Map resolution range (Å) | ≥8 | ≥9 | ≥13 |
| Refinement | | | |
| Initial model used (PDB code) | 4B03, 5AZE, 4JZN | 4B03 | NA |
| Model resolution (Å) | 9.1 | 9.4 | NA |
| FSC threshold 0.5 | | | |
| Model resolution range (Å) | ≥9.1 | ≥9.4 | NA |
| Map sharpening B factor (Å$^2$) | −900 | −912 | |
| Model composition | | | NA |
| Non-hydrogen atoms | 2642 | 1998 | |
| Protein residues | 2642 | 1998 | |
| Ligands | - | - | |
| B factors (Å$^2$) | NA | NA | NA |
| Protein | | | |
| Ligand | | | |
| R.m.s. deviations | NA | NA | NA |
| Bond lengths (Å) | | | |
| Bond angles (°) | | | |
| Validation | | | NA |
| MolProbity score | 2.15 | 1.94 | |
| Clashscore | 2.27 | 1.00 | |
| Poor rotamers (%) | NA | NA | |
| Ramachandran plot | NA | NA | NA |
| Favored (%) | | | |
| Allowed (%) | | | |
| Disallowed (%) | | | |

*SPA* single-particle analysis, *StA* subtomogram averaging

real-space refinement was subsequently employed to ensure realistic bond lengths and geometry (Fig. 3a and Supplementary Movie 1). The correlation coefficient between density calculated from the fitted capsid protein model and the corresponding density in the cryoEM map is 0.86, and root-mean-square deviation (RMSD) of the final fitted model to the ZIKV crystal structure is 2.27 Å (Supplementary Table 1 and Supplementary Fig. 7a). FSC calculation between the capsid density map and the map generated from its fitted Cα chain model (see Methods for details) showed that, at cutoff 0.5, the maps are correlated to 10.7 Å (Supplementary Table 2 and Supplementary Fig. 3d). Structural variation of the cryoEM capsid structure from the crystal structure of recombinant capsid protein is possible, as the helices are connected by flexible loops. In addition, the crystal structure of the capsid proteins were determined in the absence of lipids and RNA, and therefore may not adopt the same structure as in the virion. For example, it is well known that fitting crystal structures of isolated E proteins into the cryoEM maps of whole virions requires that domain positions are adjusted, likely due to the flexible inter-domain hinge[18]. The final structure makes biological sense, as the hydrophobic helix α1 is facing the viral lipid membrane and the positively charged helix α4 towards the RNA genome. The fit largely preserves the overall zika capsid

crystal structure (Supplementary Fig. 7a) and has a good correlation score with the density.

**Fab DV62.5 stabilizes the immZIKV particle.** The Fab DV62.5: immZIKV complexed structure showed Fab DV62.5 binds across the pr portion of the prM and the fusion loop of the E protein (Supplementary Fig. 8a-b). The equally strong densities of the red prM–E molecule and the variable regions of the Fabs (Fig. 2d and Supplementary Fig. 9) suggest the Fab binds to this position with full occupancy. On the other hand, the epitope on the blue prM–E molecule is completely concealed by the neighboring red and green prM–E complexes, and therefore no Fab was detected (Fig. 2d). Although the epitopes on the three green prM–E molecules surrounding the threefold vertex are completely exposed (Fig. 2d), the Fab densities are weaker (Supplementary Fig. 9), suggesting partial occupancy. Localized reconstruction of the densities around the threefold vertices showed two major classes of Fab binding (Fig. 2e). In the first class, there was one Fab bound to a prM–E complex near this vertex (Fig. 2e, left panel), whereas in the other class there were two Fab molecules each bound to a prM–E complex (Fig. 2e, right panel). This binding may limit motions of these prM–E spikes at this vertex.

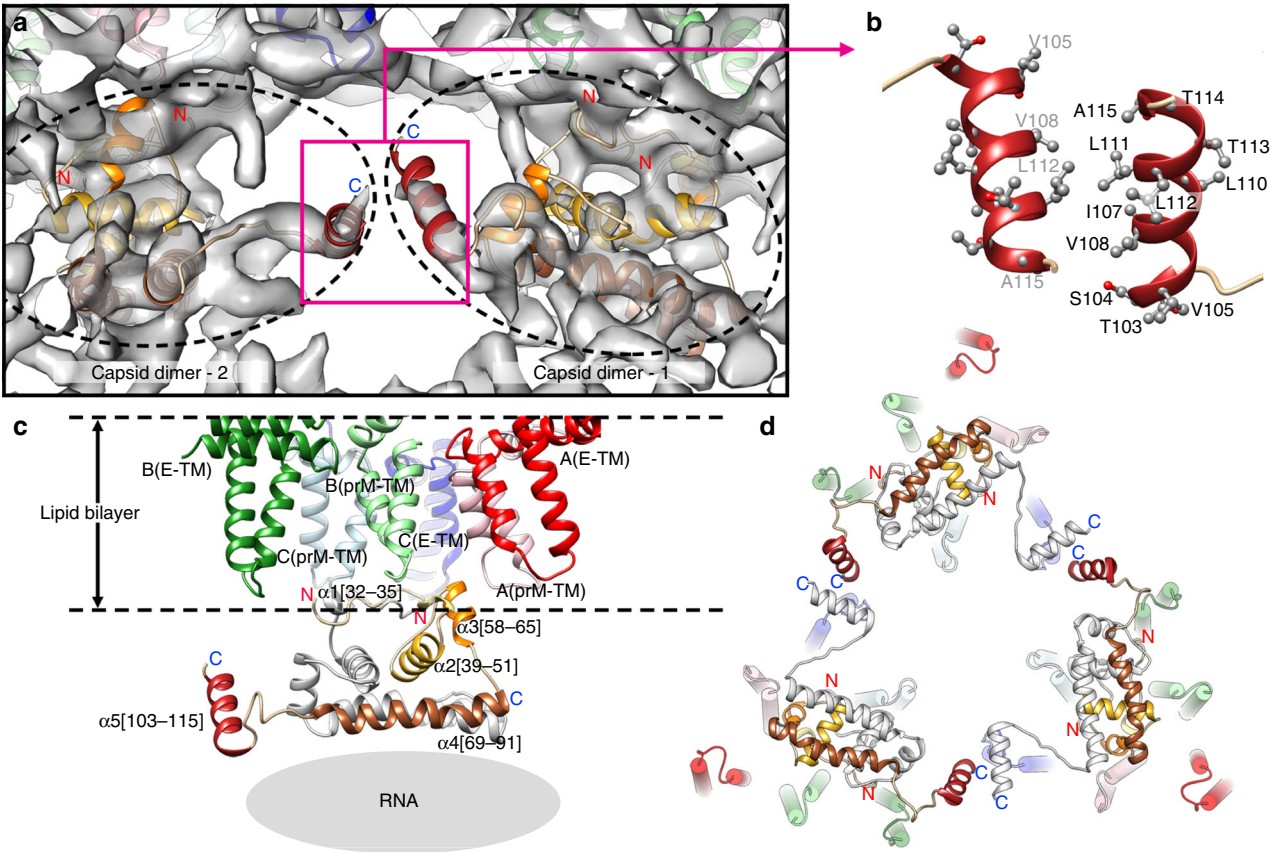

**Fig. 3 The helices α5 are important for facilitating trimerization of capsid dimers. a** The fit of crystal structure of ZIKV capsid protein dimers (dotted black circles) into the immZIKV density map (transparent gray). **b** Two capsid dimers interact via their hydrophobic interactions between helices α5. **c** Side view showing the orientation of the capsid protein with respect to the lipid bilayer membrane and the viral RNA. The capsid dimer is located below the cluster of the prM and E-TM regions. One capsid protein contains five helices (α1–α5). The helices of one capsid protomer within the dimer are colored from the lightest to the darkest shade of brown, whereas the other capsid protomer is colored in light gray. The helix α1 of both protomers clustered together forming a largely hydrophobic surface interacting with the viral lipid membrane. The helix α4 containing highly positively charged residues facing the negatively charged RNA. **d** View from the inside of the virus, three capsid protein dimers interact with each other via helix α5 forming a triangular network.

In conclusion, the combination of Fab DV62.5 binding across prM and E, and also the Fab simply occupying space on the virus surface, likely helps stabilize the overall structure.

**Capsid protein tertiary structure in ZIKV.** The surface of the inner leaflet of the bilayer lipid membrane consists of negatively charged phosphate heads. The highly negatively charged viral RNA genome would thus seem to repel the lipid surface, creating a gap between these two layers. The capsid protein exists as overall positively charged dimers (Fig. 2c, two right panels) that bridge the RNA and the lipid membrane surfaces (Fig. 2a, left bottom panel). The capsid dimers are located directly beneath clusters of the TM regions of the prM and E proteins (Fig. 3c and Supplementary Movie 3). There are 60 copies of capsid dimers in total (120 copies of capsid protein) in the virus particle. Comparison of our cryoEM ZIKV capsid structure with the NMR DENV, crystal WNV, and crystal ZIKV structures showed consistent three layers mostly helical structures (Supplementary Fig. 7b), with RMSD values of 3.31 Å, 2.53 Å, and 2.27 Å, respectively (Supplementary Table 1). Although the NMR and crystal structures of the capsid proteins were not determined in the presence of lipid and RNA, our cryoEM structure of capsid dimer in the virus particle (Fig. 3c) showed helix α1 form the first layer interacting with the inner leaflet membrane, the second layer containing helices α2 and α3, and the third layer consists of

the positively charged helix α4 that interact with the negatively charged RNA genome. Based on the position of helix α1, both our cryoEM ZIKV capsid and the crystal ZIKV capsid[6,7] structures are closer to that of the WNV capsid crystal structure (Supplementary Fig. 7b). The N-terminal end of the capsid protein, which consists of 20 residues preceding helix α1, is disordered in all NMR and crystal flavivirus capsid protein structures, and also in our cryoEM structure.

In all NMR and crystal structures of the recombinant flavivirus capsid protein[4–7], only the residues from the N-terminal end to the end of helix α4 were expressed, as it was assumed to be the mature form of the capsid, i.e., after cleavage of α5 by NS2B/NS3 protease during processing of the single translated viral polyprotein (Fig. 1a). However, in our virus structure, we clearly observed density corresponding to helix α5 in addition to density for helices α1–α4 (Fig. 3a).

**Biochemical analysis of the capsid protein.** SDS-polyacrylamide gel electrophoresis (PAGE) analysis of the immZIKV preparation showed the presence of E, prM, and capsid proteins (Supplementary Fig. 10a). There is no band that corresponds to M protein (8415 Da), suggesting the virus prep is highly immature. To separate the capsid proteins with and without helix α5, we ran the virus samples several times with a higher percentage (15%) SDS-PAGE (Supplementary Fig. 10b). There were two capsid

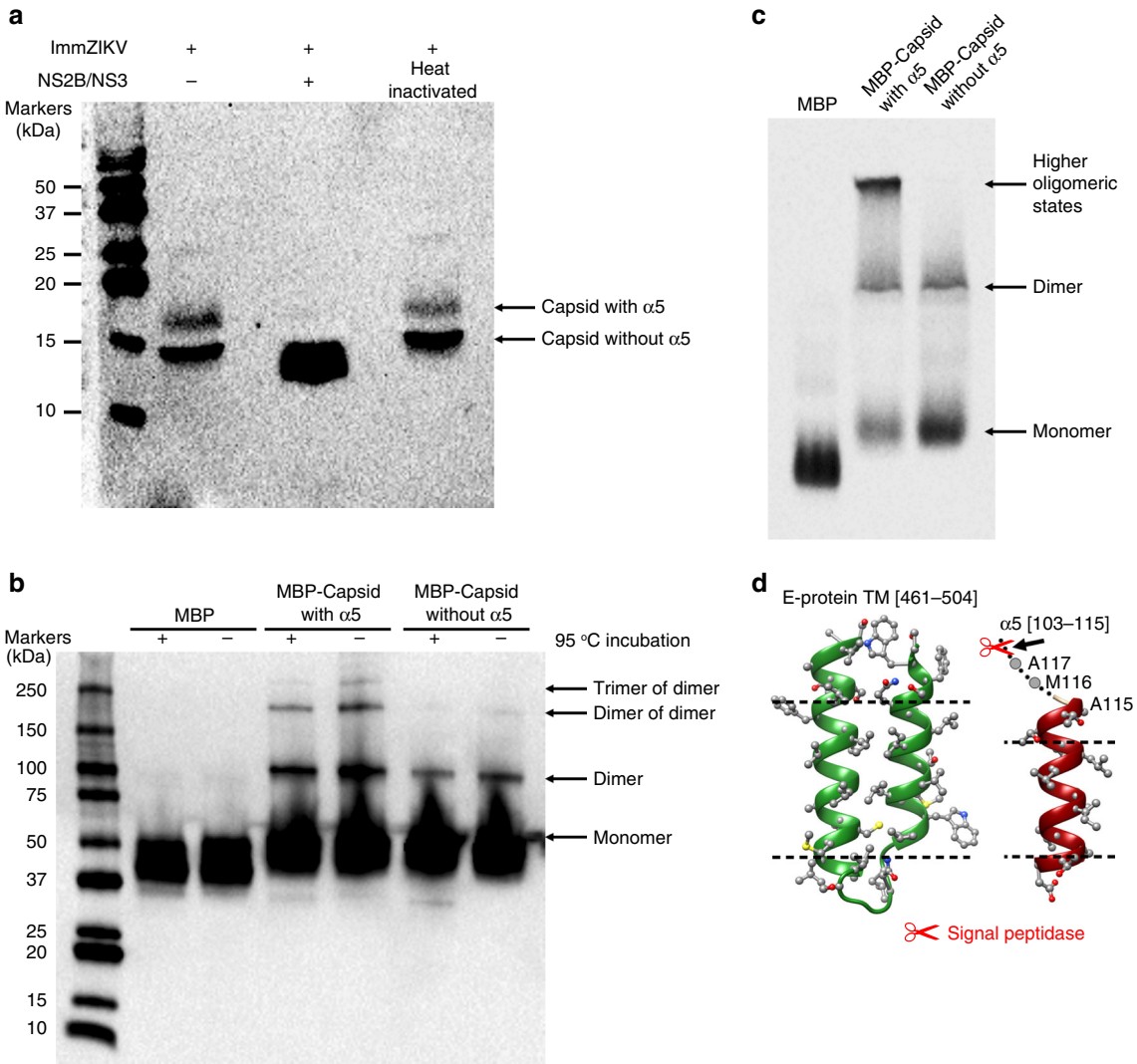

**Fig. 4 Presence of helix α5 and its importance on capsid protein dimers oligomerization. a** The higher MW capsid protein contains helix α5. The viral protease NS2B/NS3 is known to cleave full-length capsid protein between helix α4 and α5. Tartrate-purified ImmZIKV was lysed with 1% DDM and incubated with native or heat-inactivated NS2B/NS3 followed by western blotting analysis with anti-ZIKV capsid antibody. In the absence of or with heat-inactivated NS2B/NS3, two capsid bands were detected, suggesting two population of capsid proteins—likely capsid with and without helix α5. In the presence of NS2B/NS3, the intensity of the higher molecular weight capsid band decreased, whereas that of the lower band increased, suggesting that the upper band contains helix α5. **b** Helix α5 is essential for the trimerization of capsid protein dimers. SDS-PAGE western blot analysis of boiled ( + ) or not boiled ( − ) samples containing purified MBP only, MBP fused to capsid protein with and without helix α5, using an anti-MBP antibody. MBP:capsid protein with helix α5 shows the presence of dimer of dimers and trimer of dimers, and when helix α5 is absent the capsid protein is largely unable to form these higher oligomerization states. **c** Native PAGE analysis of MBP-fused capsid proteins showed that the full-length capsid proteins with helix α5 can form a higher oligomeric band but not the capsid proteins without helix α5. **d** Capsid helix α5 may not be stable in the ER membrane. Comparison of the TM region of the E protein (left) with the capsid helix α5 (right). The dotted line indicates the boundary of the hydrophobic region. This shows that the helix α5 is much shorter than that of the E-TM region, suggesting it is likely to be less stable in the ER membrane and thus could be pulled out of the membrane when the capsid protein interacts with RNA. Signal peptidase cleavage site is indicated.

bands present as stained by anti-capsid antibody in a western blotting (Fig. 4a and Supplementary Fig. 11a, b).

To detect helix α5 requires analyzing the molecular weight (MW) of whole intact capsid proteins, as the overall charge of the capsid protein is positive and hence can be analyzed in a matrix-assisted laser desorption/ionization time-of-flight (MALDI-TOF) mass spectrometry. We conducted qualitative mass spectrometry analysis of the purified whole virus. Acetonitrile (ACN) and trifluoroacetic acid (TFA) was added to dissolve the viral membrane. However, as the TM regions of E and prM are highly hydrophobic, they would interact with each other and precipitate out of solution in the absence of detergent. All

membrane-anchored viral surface proteins E (MW of 54,267 Da) and prM (18,695 Da) are thus not detected in the MALDI-TOF mass spectrometry (Supplementary Fig. 12). The inability to detect membrane-anchored proteins in MALDI-TOF had also been previously demonstrated for Sindbis virus when prepared in a similar way[19]. There may be some very low levels of contaminating mature virus particles in the virus preparation not detectable by SDS-PAGE. In MALDI-TOF, we also did not observe any M protein (8415 Da). We observed only two peaks with MW of 11,694 Da and 14,947 Da. There are no other structural proteins other than capsid protein with a MW around these peaks (Supplementary Table 3), and as their MW

corresponds closely to the calculated capsid protein without and with helix α5 (Supplementary Fig. 12 and Supplementary Table 3), this result suggests that there is partial NS2B/NS3 protease processing of capsid protein during virus assembly.

To confirm the presence of helix α5 in the capsid proteins, we added NS2B/NS3 protease to the detergent-lysed and RNase-treated immZIKV, and then detected changes in the two capsid bands in a western blotting using anti-capsid antibody. When NS2B/NS3 protease was added, the intensity of the upper band decreased whereas the lower band increased compared with virus sample with no added NS2B/NS3 protease (Fig. 4a and Supplementary Fig. 11a,b). This strongly suggests that the upper capsid band comprised capsid protein with helix α5.

**Capsid protein quaternary organization in the immZIKV.** We observed a triangular network of capsid dimers (Fig. 3d)—three capsid dimers interact with each other via their helix α5 (Fig. 3a, d). This likely occurs via hydrophobic interactions between the helices α5 (Fig. 3b).

The detection of capsid protein containing helices α1–α4 by mass spectrometry suggests that these triangular networks of capsid dimers may contain some capsids without helix α5; thus, the observed helix α5 densities are likely a result of averaging densities of capsid proteins with and without helix α5. This is supported by the observation of slightly weaker densities of helix α5 compared to the other helices. The lack of icosahedral symmetry of the viral RNA genome precludes observing whether there are other capsid proteins interacting with the RNA genome that are not at this lipid envelope:RNA interface. It is therefore also possible that the capsid proteins with helix α5 are concentrated at the lipid envelope:RNA interface, i.e., if trimerization of the capsid dimers are important for interaction with the TM regions of surface prM and E proteins, whereas those without helix α5 are at the inner core of the RNA.

We observed each capsid dimer is located in close proximity (<12 Å distance between the pairs of Cα backbone) to three prM-TM and one E-TM (green E protein) regions (Fig. 5a, top right panel), indicating possible interactions. Examination of the residues at these interfaces shows complementary charges (Supplementary Fig. 13). Possible interactions are as follows: (1) the capsid N-terminal loop preceding helix α1 and also helices α1 and α3 interacting with the loop that links the two anti-parallel TM helices (Supplementary Fig. 13b-d) of the three individual prM-TMs, and (2) the capsid N-terminal loop with the green E-TM region (Supplementary Fig. 13a). These interactions would thus cluster the prM proteins on the virus surface. Our current cryoEM immZIKV structure, the previously solved DENV prM–E crystal structure[20], and the cryoEM structure of the immDENV[14] showed extensive interactions between the prM and E protein on the surface of the virus (Supplementary Fig. 1a). This suggests that the capsid protein dimer may thus interact with the prM molecules of these preformed prM–E complexes (Fig. 5a and Supplementary Movie 3). The preformed nine E–prM heterodimers with a trimer of these capsid dimers at the threefold vertices (Fig. 5b) therefore forms an assembly unit (Fig. 5c). The tips of these prM–E complexes in one assembly unit on the outer viral surface then interact with other assembly unit (Supplementary Fig. 1c,d), forming the overall virus quaternary structure (Supplementary Movie 3).

To determine ability of helix α5 to induce capsid dimers to form higher oligomers, we purified maltose-binding protein (MBP)-tagged ZIKV capsid proteins with or without helix α5. In Coomassie-stained SDS-PAGE gels (Supplementary Fig. 10c), all samples, whether boiled or not boiled, showed only the monomeric form. On the other hand, in western blottings using

an anti-MBP antibody, we detect some bands of the MBP:capsid protein with helix α5 that correspond to its higher oligomerization states (Fig. 4b)—dimer, dimers of dimers, and trimers of dimers. By contrast, the MBP:capsid protein without helix α5 showed largely the absence of dimers of dimers and trimers of dimers. We also ran the MPB:capsid protein with and without helix α5 on a native PAGE and the results showed that in the presence of helix α5, a higher oligomer capsid was detected (Fig. 4c). As the helix α5 is amphipathic in nature, it is not surprising that helix α5 in capsid dimers can interact with each other to form higher oligomers. The asymmetric tomogram-averaged map suggests that in some icosahedral equivalent positions, there are missing capsid dimers in the proposed triangular network; however, if capsid dimers with helix α5 is located close to another capsid dimer, they likely interact via their helix α5 as shown in our icosahedral-averaged SPA map.

**Discussion**

It is known that after translation of the viral polyprotein, the capsid remains on the cytoplasmic side of the ER and its helix α5 transverses through the ER membrane leading to the translocation of prM into the ER lumen (Fig. 1a) (reviewed by Byk and Gamarnik)[21]. At the N- and C-terminal ends of helix α5 are the NS2B/NS3 protease and the signal peptidase cleavage sites, respectively. It has been shown that there is competition between the signal peptidase and the NS2B/NS3 protease to reach their respective ends of the helix α5, which maybe partially hidden on either side of the ER membrane, thus leading to sequential cleavage[22]. Regardless of whether the signal peptidase or NS2B/NS3 cleave first, it was proposed that the final product is (1) the final mature capsid protein containing residues from the N terminus to the end of helix α4 and (2) the helix α5 being cleaved off from the N terminus of the prM molecule. However, our observation that the cryoEM structure clearly shows densities of helix α5, combined with MALDI-TOF analysis of purified virus particles and also NS2B/NS3 digestion of the viral capsid proteins, confirm the presence of a mixture of capsid proteins with and without helix α5. The presence of helix α5 in capsid proteins assembled in virions suggests that cleavage by NS2B/NS3 could be incomplete or partially inhibited. As we observed the helix α5 in the capsid protein layer in the virus, it suggests that this previously membrane-associated helix has been pulled out of the ER membrane. Indeed, capsid helix α5 is shorter than the TM regions of the E protein (Fig. 4d) and also the other typical TM helix[23]. In a typical TM helix, ~17.7 residues span the hydrophobic part of the lipid bilayer, whereas in capsid helix α5, likely only 8 of the 13 residues are involved. The capsid helix α5 also has a lower percentage of hydrophobic residues compared with a typical TM helix (53.9% vs. ~78.5%). We also used a website (http://dgpred. cbr.su.se/index.php?p=instructions) to calculate the $\Delta G$ for TM helix insertion for capsid helix α5 into the ER membrane by means of Sec61 translocon[24,25]. A negative value would suggests the helix is able to be efficiently integrated into the membrane, whereas a positive value suggests poor efficiency and its insertion may require other stabilizing interactions. The calculation shows a $\Delta G^{pred}_{app}$ value of +2.45, suggesting capsid helix α5 may not readily transverse through the ER membrane. Although the capsid helix α5 must transverse the membrane during initial polypeptide processing, the above arguments suggest its predicted length and residue properties may not allow it to be stably integrated into the ER membrane.

MALDI-TOF determination of the capsid protein with helix α5 shows a total MW of 14,947 Da, which accounts for all capsid protein residues (13,391 Da) and an additional ~1500 Da (Supplementary Table 3). As the hydrophobic helix α5 must be

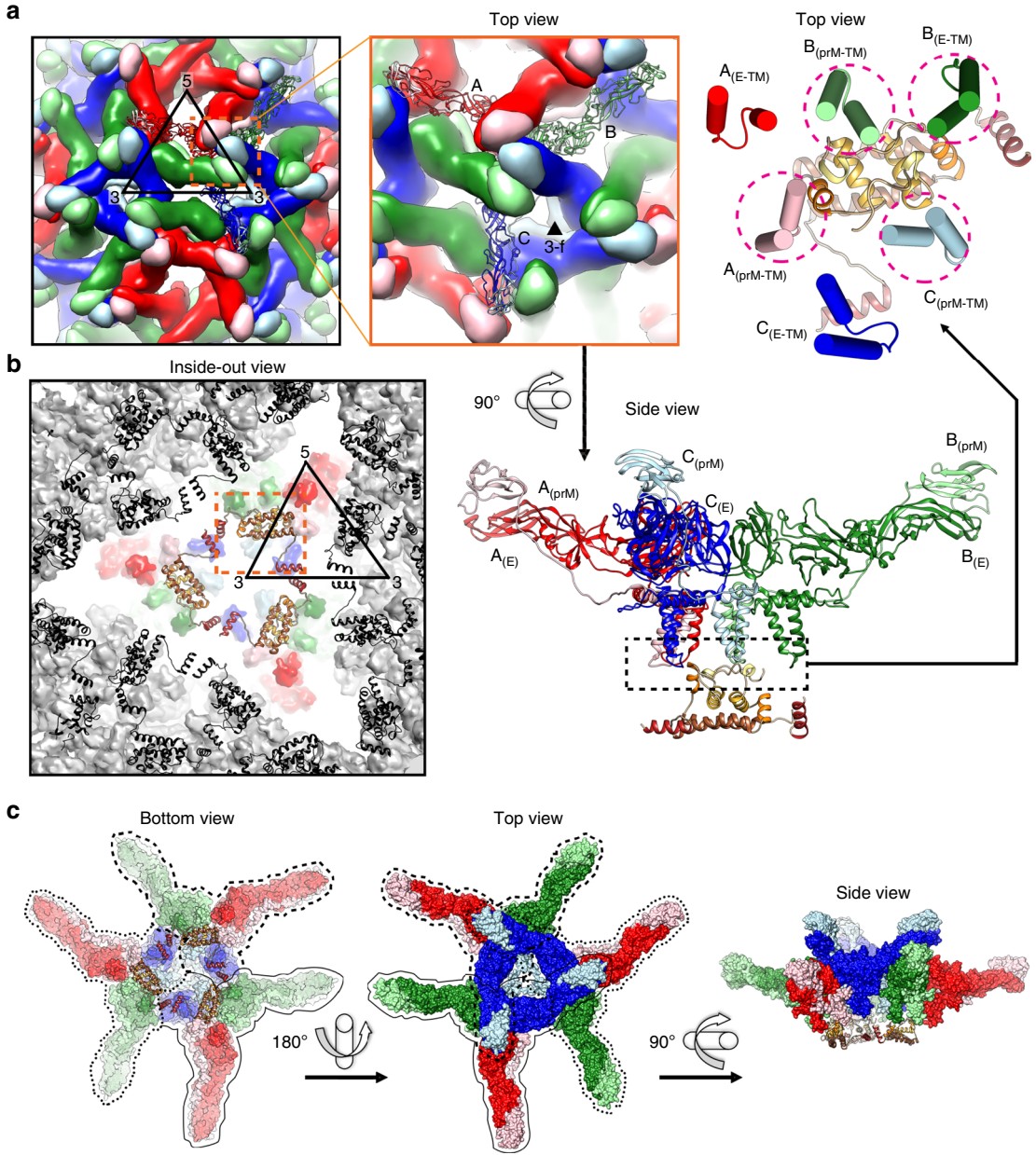

**Fig. 5 Capsid protein is important for the organization of the virus quaternary structure. a** Capsid protein dimer is located in between the fivefold and threefold vertices (left and middle top panels); it acts as a base supporting the red, green, and blue prM–E-protein complexes (bottom panel). Right-top panel, one capsid dimer likely interacts with the loop that links the two anti-parallel helices of the TM regions of three prM proteins and the green E protein (pink dotted circles). **b** View from the inside of the virus, three capsid protein dimers form a triangular network around the threefold vertices. Neighboring capsid dimer triangular networks around other threefold vertices are colored in black and the prM–E complexes as gray surfaces. The dotted orange box indicates the same location as in **a**, but viewed from the inside of the virus. **c** Each capsid protein dimer binds to three E–prM protein complexes (circled in dotted or dash, or black lines); therefore, one triangular network of capsid protein dimers brings together a total of nine E–prM heterodimers, forming an assembly unit.

threaded through the ER membrane during viral polyprotein processing before subsequently being pulled out of the membrane, the extra ~1500 Da may correspond to two copies of phospholipids (MWs of phosphoethanolamine or phosphatidylcholine are about 690 or 760 Da, respectively), which may have been pulled out along with the hydrophobic helix α5. Alternatively, the extra ~1500 Da could be due to some covalent modifications such as phosphorylation and/or myristoylation of the capsid protein with helix α5.

The ability to make virus-like particles (VLPs) by expressing only prM and E proteins (in the absence of capsid proteins)

suggests a more active role of these proteins in the virus assembly process[26]. However, secreted VLPs have irregular sizes and most are smaller (predominantly particles containing only 60 copies of prM–E) than the native virus, suggesting the viral assembly involving only E and prM proteins is unable to produce typical flavivirus icosahedral particles containing 180 copies of prM–E. The capsid protein therefore may serve as an important mediator between the surface glycoproteins and viral RNA, ensuring the virus is packaged properly. However, at the same time, as shown in our asymmetric subtomogram-averaged map (Supplementary Movie 2), capsid dimer occupancies are not full, suggesting that

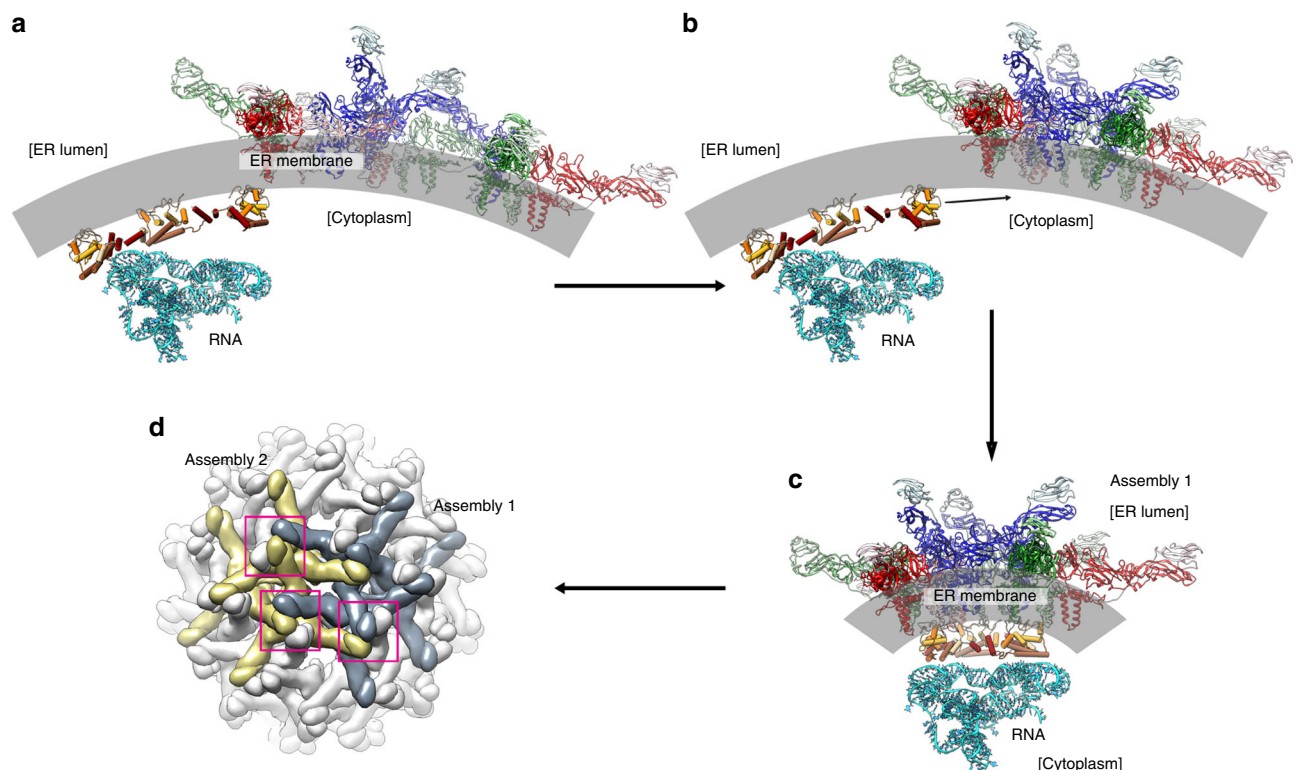

**Fig. 6 Proposed assembly process of immZIKV particle. a** Capsid proteins are expressed in the cytoplasmic side of the ER membrane, where they form homodimers. One side of the homodimer interacts with ER membrane, whereas the other side with viral RNA. Simultaneously, on the luminal side of the ER, the prM and E proteins form heterodimers and then three of these heterodimers associate with each other to form an inverted tripod. **b** The inverted tripods interact with each other. The capsid dimers then attach to the TM regions of the prM/E tripods. **c** The capsid dimers then interact with other nearby capsid dimers via their helix α5. This sets the spacing between the inverted tripods. Three prM/E inverted tripods with capsid protein dimers binding underneath form an assembly unit. **d** The tips of these assembly units (pink boxes) then interact to form a virus surface lattice on the lumen side of the ER membrane. One assembly unit is colored in gray and the other in yellow.

the maintenance of the trimeric interactions of the capsid dimers at all icosahedral equivalent sites is not strictly necessary for assembly. Here we propose a series of events that may occur during the virus assembly process (Fig. 6) as follows: (1) after the virus polyprotein is translated, the capsid proteins on the cytoplasmic side of the ER form homodimers (Fig. 6a). Simultaneously, on the luminal side of the ER, the prM and E proteins form heterodimers and then three of these heterodimers associate with each other to form an inverted tripod (Fig. 6a). These prM/E inverted tripods then interact with each other (Fig. 6b). (2) On the ER cytoplasmic side, some of the capsid dimers interact with the TM hairpins of three inverted prM/E tripods. The capsid dimers, which are in close proximity to each other, will make lateral interactions via their helix α5, further strengthening the assembly (Fig. 6c). This forms an assembly unit. (3) The tips of the inverted prM/E tripods from each assembly unit will interact with each other at the luminal side of the ER creating a lattice on the ER membrane (Fig. 6d). The function of capsid protein dimers could be to provide the correct spacing between the prM–E inverted tripods. The prM/E inverted tripods thus attract the capsid protein:RNA complex to the budding site.

There is a recent publication of asymmetric SPA reconstructions of the uncomplexed immature Kunjin and ZIKV viruses[27]. Although the maps are too low in resolution (20 Å) to show the capsid protein layer, they observed a blob of RNA genome density on the inside of the particle that suggests the RNA genome may be located slightly off-centered after the virus has assembled in the ER.

During the maturation process, the E and prM proteins on the virus surface undergo dramatic structural rearrangements, and

their start and end positions had been determined previously[14] (Supplementary Fig. 14). Capsid protein densities are observed in our icosahedral reconstruction of the immature virus, while those in the mature virus are largely absent[10,11]. In our immature virus structure, one capsid dimer likely interacts with three prM-TM regions (Supplementary Fig. 14c, left panel). Comparison of the positions of prM-TM in the immature particle with those in the mature virus (M-TM) (Supplementary Fig. 14c, right panel) showed that the prM-TM dramatically rearranged from a trimeric structure to a dimeric one[14]. Thus, our observed capsid dimer interaction with the three prM-TMs in the immature virus would be broken during the maturation process. It is possible that this would result in the capsid–RNA complex completely or partially detaching from the M-TMs in the mature virus. This may aid the virus to release its capsid protein into the cell cytoplasm in the next cycle of infection after fusion with the endosomal membrane. Alternatively, the capsid dimer may follow one of the three M-TMs. Hence, there would be three different possible capsid positions in any of the 60 asu within the mature virus, leading to weaker density during icosahedral symmetry averaging (Supplementary Fig. 14c). In addition, the weakened interaction due to capsid binding to only one prM-TM instead of three will likely result in increased motions of the protein, which will further degrade observable densities.

The structure of the capsid protein in the virus particles indicates that it plays an important role in the virus assembly process. It also illuminates key interactions with other viral structural protein components, thus providing new targets for drug design.

## Methods

**Culture and purification of immZIKV.** C6/36 cells (American Type Culture Collection) were grown to 90% confluency in a ten-cell stack. The cells were then infected with ZIKV (H/PF/2013) at multiplicity of infection of 1 for 16 h at 29 °C. After infection, the inoculum was discarded and infected cells were washed three times with phosphate-buffered saline. To minimize mature virus contamination, minimum essential medium (MEM) containing 2% fetal bovine serum (FBS) and 30 mM ammonium chloride were added to infected cells and incubated for 1.5 h at 29 °C. This process was repeated two more times before adding fresh MEM supplemented with 2% FBS and 30 mM ammonium chloride. At 48 h post infection, cell culture supernatant was collected and centrifuged at 8983 × g for 30 min, to remove cell debris. For purification of immZIKV, the virus in supernatant was precipitated overnight with 8% (w/v) polyethylene glycol 8000 and pelleted the following day at 8900 × g for 50 min at 4 °C. Pelleted viruses were resuspended in NTE buffer (10 mM Tris-HCl, 120 mM NaCl, 1 mM EDTA pH 8) and subjected to further purification through a 24% sucrose cushion, before finally going through a linear 10–30% (w/v) potassium tartrate gradient. A visible virus containing band was extracted and buffer exchanged into NTE buffer with the Amicon Ultra-4 100 kDa concentrator (Millipore). A small aliquot of purified virus along with a series of bovine serum albumin standards were analyzed on SDS-PAGE with Coomassie blue staining, to determine the purity and concentration of the immZIKV preparation.

**CryoEM sample preparation.** ImmZIKV-Fab DV62.5 complex sample was prepared by mixing purified immZIKV with Fab DV62.5 at an E protein-Fab DV62.5 ratio of 1:1.1 and then the mixture was incubated at 37 °C for 30 min. A 2.3 µL sample of virus:antibody complex or uncomplexed immature virus was applied onto glow-discharged ultra-thin lacy carbon grids (Ted Pella) and blotted with filter paper before plunge freezing in liquid ethane using the Vitrobot Mark IV (FEI). Frozen grids were stored in liquid nitrogen temperature.

**CryoEM data collection, image processing, and reconstruction.** Vitrified viruses were imaged using a 300 kV Titan Krios microscope (FEI) fitted with a Falcon II direct electron detector (FEI). Automated image data collection was carried out using Leginon software[28,29]. For uncomplexed immZIKV, 3366 micrographs at ×59,000 magnification with pixel size of 1.34 Å were collected (Table 1). For immZIKV-Fab DV62.5 complex, 2323 micrographs at ×47,000 magnification with pixel size of 1.71 Å were collected (Table 1). Contrast transfer function parameters were estimated using the GCTF program[30]. Micrographs showing significant astigmatism and drift were discarded. Only micrographs within the defocus range of 0.5–3.5 µm were included for further processing. For each image dataset, initial particle picking was done manually for ~1000 particles using the e2boxer function in EMAN2 (ref. [31]) and then two-dimensional (2D) classification was done on these particles using Relion2.1 (ref. [32]). Representative 2D class averages derived from manually picked particles were used as reference for automated particle picking using the Gautomatch software (downloaded from http://www.mrc-lmb.cam.ac.uk/kzhang/). Further cryoEM reconstruction processing was done using Relion2.1. Two rounds of 2D classification were carried out to remove broken particles. Three-dimension (3D) classification with particle alignment and icosahedral symmetry imposed was performed using the cryoEM map of immZIKV (EMD-8508) as a starting model. Particles in good 3D class averages were combined and final 3D reconstruction was done. The final cryoEM maps were subjected to B-factor sharpening in Relion2.1. In total, 7922 particles were selected in the reconstruction of the uncomplexed immZIKV, while 19,295 particles were selected to reconstruct the immZIKV-Fab DV62.5 complex. The resolution of each map was estimated using 0.143 cutoff the gold standard Fourier shell correlation curve calculated from two maps that were reconstructed from independent two-half datasets. All FSC calculations in Fig. 2 were done using EMAN[33]. In Supplementary Fig. 3, all FSC calculations are repeated using Phenix mtriage—the resolution determined for these maps are similar between programs. For resolution estimation for the capsid and the E ectodomain layers in the uncomplexed immZIKV and the immZIKV:DV62.5 Fab complex maps, we applied a soft-edge radial shell mask either covering capsid or pr-E ectodomain densities layer.

**Localized reconstruction.** Localized reconstruction of the densities around the threefold vertices was done according to the published protocol by Ilca et al.[34]. Briefly, the locations of the projections of the densities at threefold vertices on each of the particle images were located based on the orientations of the particles used in the final cryoEM reconstruction step mentioned above. These individual projections were then extracted from each particles to make subparticle images and the CTF parameters were adjusted according to the location of the subparticles on the particle. The 3D classification without alignment was performed with C3 symmetry applied by using Relion. One of the class that contained 255,170 subparticles showed better densities with higher resolution compared to other classes. Further 3D classification without alignment was done with C1 symmetry and a mask that covering the densities correspond to the Fab DV62.6 was applied. The mask was created by simulating the densities of three Fab molecules that bind simultaneously to the three green prM–E proteins located adjacent to threefold vertex in Chimera followed by mask creation that was done in Relion with additional 8 pixel map extension and 16 pixel soft-edge. Two major classes of subparticles were found. The

first class (62,058 subparticles) showed 1 Fab DV62.5 bound to one of the green prM–E proteins adjacent to threefold vertices (Fig. 2e, left panel). The second class (171,006 subparticles) showed two Fab molecules bound to two adjacent green prM–E proteins (Fig. 2e, right panel).

**CryoEM tomography data collection and image processing.** A sample of ImmZIKV-Fab DV62.5 was mixed with solution of 10 nm gold particles, put on a thin carbon on lacey carbon-coated copper grids, blotted, and flash-frozen in liquid ethane. Tilt series were collected at ×47,000 nominal magnification (pixel size 1.71 Å), at an underfocus of 3–4 µm (Table 1). The sample was tilted in sequence from 0° to −30°, then 0° to 60° with 3° step, with a total dose of 90 e−/Å². Images were collected in movie mode and the frames were combined using MotionCor software[35]. The tilt series were processed with IMOD[36]. In total, 63 tilt series were collected and processed.

**CryoEM subtomogram averaging.** Individual virus particle tomograms were selected with template matching procedure in emClarity[37], using a ×10 down-sampled 3D map of an icosahedrally averaged immDENV1 cryoEM structure as a search template. A total of 1934 individual particle tomograms were selected. Subtomogram averaging was carried out in emClarity, with no symmetry applied. After ten cycles of refinement in emClarity, the resolution for the subtomogram-averaged map had reached 16 Å. Then, an additional selection of tomograms was done for further refinement. This selection involved first icosahedrally averaging the individual subtomograms with their pre-determined orientation parameters and then compared with the icosahedrally averaged model using FSC. Particle tomograms (1152) showing resolution of better than 40 Å at FSC cutoff of 0.5 were selected for further refinement. After another five cycles of refinement using these subtomograms with no symmetry applied, the final averaged map was determined to 13 Å resolution (reported by emClarity).

**Fitting of prM–E and capsid structure into the density maps.** The model building in the cryoEM maps of uncomplexed immZIKV and immZIKV-Fab DV62.5 complex were done by fitting the crystal structure of ZIKV capsid protein (PDB ID: 5YGH) and homology models of E protein, prM, and heavy and light chains of Fab DV62.5 that were prepared separately using Swiss-model server[38]. The immature DENV serotype 1 structure (PDB ID: 4B03) was used as template to build the homology model of immZIKV E and prM proteins, whereas two human antibody structures (PDB IDs: 5AZE and 4JZN) were used as template for the heavy and light chains of Fab DV62.5, respectively. The fitting of E and prM proteins into the cryoEM maps of immZIKV-Fab DV62.5 complex and uncomplexed immZIKV were done using the "Fit in Map" tool in Chimera program[39]. The Fab DV62.5 model was also fitted into the density in the cryoEM map of immZIKV-Fab DV62.5 complex using Chimera. There are two possible orientations of the Fab molecules in the density, but one of the two fits resulted in a higher correlation coefficient (0.92 vs. 0.87). Surface charge analysis of the interacting surfaces in the final fitted structure showed charge complementarity between the Fab paratope and the epitope (Supplementary Fig. 8c). The surface charge analysis was done in Chimera. The model building of capsid protein into the immZIKV-Fab DV62.5 complex maps calculated with different level of sharpening (B factors of −900 and −550 Å²) was done manually using the program O[40]. The capsid protein model was initially fitted into the density as a rigid body dimer and then the fit was optimized by fitting individual helices separately. Further model building and structure regularization was done in COOT[41]. The prM/E and capsid protein models fitting into uncomplexed immZIKV and immZIKV-Fab DV62.5 complex were refined using the "Real-space refinement" tool[42] in the Phenix program[43] with the secondary structure restraint applied resulting in the final overall correlation coefficient of 0.80 and 0.76, respectively. The correlation coefficient between the density calculated from the fitted capsid model and the corresponding density in the cryoEM map showed a value of 0.86. This was determined by using the Fit in Map function in Chimera to calculate the correlation coefficient between a simulated 9.5 Å resolution map generated from capsid protein model and the corresponding cryoEM density. The program Phenix.mtriage[44] was used to calculate the correlation of the fitted protein models to the cryoEM map. We extracted the densities of capsid protein and pr-E proteins from the icosahedrally averaged SPA map and then Phenix mtriage generated the model maps from Cα backbone of fitted molecules and conducted FSC calculation. A soft mask was calculated using default settings in Phenix.mtriage[44]. At a cutoff of 0.5, the models correlated to 10.7 and 9.1 Å, respectively (Supplementary Tabel 2 and Supplementary Fig. 3c,d). Structural alignment of our cryoEM ZIKV capsid structure to the NMR DENV, crystal WNV, and crystal ZIKV structures to calculate the RMSD values was done by using Superpose program[45].

**Sequence analysis.** The sequence alignments of flaviviruses was done using MultAlin, multiple sequence alignment webserver[46]. The aligned amino acid sequences were forwarded to ESPript 3.0 (ref. [47]) to render the sequence similarities (Fig. 1b). The amino acid sequences used for comparison were ZIKV strain French Polynesia H/PF/2013 (GenBank accession code AHZ13508.1), WNV strain NY99-flamingo382-99 (GenBank accession code AAF20092.2), Japanese encephalitis virus strain WHe (GenBank accession code ABL60896.1), TBEV strain Neudoerfl

(GenBank accession code AAA86870.1), Yellow fever virus strain French (GenBank accession code AAA99713.1), DENV serotype 1 strain Myanmar 49440 (GenBank accession code AEM92304.1), DENV serotype 2 strain New Guinea C (GenBank accession code AAA42941.1), DENV serotype 3 strain 05K863DK1 (GenBank accession code ABW82026.1), and DENV serotype 4 strain 06K2270DK1 (GenBank accession code ADK37472.1).

**MALDI-TOF mass spectrometry analysis of immZIKV.** The MW of intact proteins in purified immZIKV particles (~0.15 mg mL$^{-1}$ in E-protein concentration) were measured by MALDI mass spectrometry analysis. A volume of 10 μL of 100% ACN and 1 μL of 2% TFA were added to 10 μL of 0.15 mg mL$^{-1}$ immZIKV particles to produce a final mixture containing 0.0714 mg mL$^{-1}$ of immZIKV particles in 50% ACN and 0.1% TFA. The sample was subjected to centrifugation at 16,100 × $g$ for 5 min and 1 μL of the supernatant was spotted on MALDI sample plate and air-dried at room temperature. Dried sample was overlaid with 1 μL of matrix solution (α-Cyano-4-hydroxycinnamic acid, diluted in 50% ACN and 2.5% TFA, Sigma-Aldrich, Milan, Italy) and subjected to MALDI-TOF mass spectrometry analysis using 4800 MALDI-TOF/TOF$^{TM}$ (AB SCIEX, Singapore) scanning in reflectron linear mode. Spectra were generated from data deriving from 6000 single laser shots from different positions of the sample spots and collected over a range from 9000 to 80,000 Da. Before collecting the data for the samples, the instrument has been calibrated with external standards (Albumin M + H$^+$ = 66,463, Albumin M + H$^{2+}$ = 33,231) across the experimental MW range (9997–80,682 m/z). The observed average MWs (M + H$^+$) of singly and doubly charged albumin, collected over 2000 shots, were 33,233 and 66,415 m/z, respectively. This matched very well with the theoretical MWs of the protein standards.

**Expression of MBP-tagged ZIKV capsid protein.** MBP and MBP-tagged ZIKV capsid proteins with or without helix α5 were expressed in *Escherichia coli*. Briefly, *E. coli* BL21 transformed with their respective expression plasmid were grown to a density of OD 0.8 at 16 °C and then protein expression was induced by addition of 0.2 mM Isopropyl β-D-1-thiogalactopyranoside, the culture were further incubated for 24 h. Cell pellets were collected by centrifugation and then resuspended in buffer, which composed of 1% DDM (Avanti), 50 mM Tris (pH 7.9), 100 mM NaCl, and 10% v/v glycerol. The cells were then sonicated and the soluble MBP-tagged capsid proteins were purified by using an amylose resin column followed by Superdex 200i column (GE Health Care).

**Western blot analysis of MBP-tagged ZIKV capsid protein.** For western blot analysis, equal amounts of purified recombinant proteins were either boiled or not boiled, were loaded and separated on a 4–20% SDS-PAGE (Bio-Rad). For native PAGE analysis, equal amounts of purified recombinant proteins were prepared in native PAGE sample prep kit (Invitrogen) and separated on 4–20% native PAGE (Bio-Rad). For both western blotting and also native PAGE, the MBP-tagged capsid proteins were detected by using a primary anti-MBP antibody (NEB, catalog number E8032S) at 1:5000 dilution followed by a horseradish peroxidase (HRP)-conjugated goat anti-mouse secondary antibody (Invitrogen, catalog number A16072) at 1:10,000 dilution.

**NS2B/NS3 protease cleavage assay.** Tartrate-purified ImmZIKV in NTE buffer was lysed in the presence of 1% DDM (Avanti) for 1 h at room temperature. To enhance viral lysis, viral samples were also sonicated for a total of 15 min. The NS2B/NS3 protease used for this assay was provided by Dr Dahai Luo (Addgene plasmid #86846; http://n2t.net/addgene:86846; RRID: Addgene_86846). Heat-inactivated NS2B/NS3 was prepared by pre-incubating the protease at 100 °C for 15 min and cooled down to room temperature prior use. Lysed ImmZIKV were then incubated with either NS2B/NS3 or its heat-inactivated form (0.08 mg mL$^{-1}$), and RNaseA (0.1 unit μl$^{-1}$) for 5 h. To look for capsid protein cleavage, the samples were then analyzed on western blotting and probed with anti-ZIKV capsid antibody (GeneTex, catalog number GTX133317) at 1:5000 dilution followed by HRP-conjugated goat anti-rabbit secondary antibody (Invitrogen, catalog number 65-6120) at 1:10,000 dilution.

**Reporting summary.** Further information on research design is available in the Nature Research Reporting Summary linked to this article.

## Data availability
The cryoEM maps of immZIKV-Fab DV62.5 complex and uncomplexed immZIKV were deposited in the Electron Microscopy Database (EMDB) under accession number EMD-0932 and EMD-0933, respectively. The coordinates of immZIKV-Fab DV62.5 complex and uncomplexed immZIKV were deposited in the Protein Data Bank (PDB) under accession codes 6LNT and 6LNU, respectively. The subtomogram-averaged map of immZIKV-Fab DV62.5 was deposited in the EMDB under accession number EMD-0934.

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

## Acknowledgements

We thank the European Virus Archive for providing us with the ZIKV H/PF/2013 strain[48]. This work is supported by Singapore Ministry of Education Tier 3 grant (MOE2012-T3-1-008), National Research Foundation Investigatorship award (NRF-NRFI2016-01), and National Research Foundation competitive Research Project grants (NRF2016NRF-CRP001-063 and NRF2017NRF-CRP001-027) awarded to S.-M.L., and the Duke-NUS Signature Research Programme funded by the Ministry of Health, Singapore. M.C.M. is supported by National Institute of Health awards R01 GM122979 and R01 GM127365, and by National Science Foundation award MCB 19227.

## Author contributions

T.Y.T. and X.-N.L. prepared samples for cryoEM studies. D.C. isolated and purified HMAb DV62.5. T.Y.T., X.-N.L., and T.-S.N. optimized cryoEM sample-freezing conditions. T.-S.N. and J.S. collected cryoEM images. T.Y.T., G.F., J.W., S.Z., and V.A.K. performed image processing and single-particle reconstructions. V.A.K performed cryoEM tomograms reconstruction and subtomo volume averaging of immZIKV particles. G.F. performed localized reconstruction. G.F. and S.-M.L. fitted capsid structure into cryoEM map. T.Y.T., G.F., V.A.K., and S.-M.L. fitted the structure. X.-X.L. performed mass spectrometry analysis. G.F. and S.-M.L. prepared the figures. S.-M.L. and M.C.M. supervised the project. T.Y.T., G.F., and S.-M.L. wrote the manuscript with support from all the authors.

## Competing interests

The authors declare no competing interests.
