## [Peer Review File · Nature Communications]

Reviewers' Comments:

Reviewer #1:

Remarks to the Author:

The manuscript by Tan et.al., uses multiple techniques to characterize the structure of the Zika virus capsid protein in the immature virion. The structure and orientation of the flavivirus capsid protein within an intact virion has long been elusive. The first indication of some order in the Zika virus capsid organization was presented by Mangala Prasad et.al in 2017. This paper provides a more substantial look into the capsid organization in immature Zika virus by using a Fab complex with the immature virus that stabilizes the capsid layer. The authors also identify that an additional helix of the flavivirus capsid protein is present in the virion which was previously not known. In addition, the authors compare the Fab DV62.5 binding to immature ZIKV and immature DENV to show that the same Fab does not stabilize the imm.DENV capsid. Some comments for the authors:

1. At an overall resolution of 8 Å and a local resolution of 9.2 Å at the capsid protein layer in their Immature ZIKV+Fab map, it is possible to fit structures into the EM map with high confidence but the statement in the abstract - 'All secondary structural elements are visible' is not entirely correct as you cannot separate 'all' secondary structural elements of the capsid protein in their current map without ambiguity.
2. The authors report a 16 Å resolution sub-tomogram averaged map of the imm.ZIKV+Fab complex to show that some capsid proteins do not have full occupancy in the virus. But there is no description of the method and parameters used in their experimental methods section. This needs to be added and explained.
3. Following up with the last comment, the supplemental figure 5 which shows the averaged sub-tomogram map of imm.ZIKV+Fab needs to be annotated more clearly to show that the capsid protein in red ribbons is in an icosahedrally equivalent position to the capsid protein in green ribbons.
4. The panel c in Figure 4 is a bit hard to follow. It would be easier if all the Fab molecules were given a similar color or texture scheme. For example, in the left most panel of Imm.ZIKV+Fab, one set of Fabs are colored in purple and have a smooth shape similar to that of the viral proteins, whereas the Fab which binds near the 3-fold vertex is in grey and has a more rough appearance. It would be useful if the authors could modify the figure so that all the Fab molecules have a distinct rough/bumpy surface so as to separate them clearly from the smoothly shaped viral proteins.
5. The authors state that the DV62.5 Fab binds less strongly to imm. DENV as compared to imm.ZIKV and point to this lower binding as the reason why the imm. DENV capsid is not stabilized by the Fab binding. As the structures of the Fab binding to DENV and ZIKV are very similar, what is the author's take on why the Fab binds less strongly to imm. DENV compared to imm. ZIKV?
6. The unliganded immature ZIKV structure has a partially ordered capsid protein layer whereas the unliganded immature DENV structure shows no such feature. Also, as the authors point out, the ZIKV capsid protein structure is more similar to WNV than DENV. Could a difference in capsid protein sequence and/or structure be responsible for the more stable organization of the capsid protein in immature ZIKV as opposed to DENV? Can the authors comment on it?

Reviewer #2:

Remarks to the Author:

In this manuscript, Tan et al describe the structure of the Zika virus core particle, determined by cryo-EM to around 9Å resolution. This is the first time the organization of the core of a flavivirus is visualized, showing that it displays icosahedral symmetry and is formed by 60 dimers of the capsid protein (or core protein, termed "C"). The authors used a non-neutralizing antibody against Zika virus that targets an epitope at the prM/E interface distal from the membrane. Other cryo-EM reconstructions of flavivirus immature and mature particles had been reported before, but in all of

them, the core was disordered. The new results are important because they open the way to understand how the viral genome is incorporated into particles during budding. The envelope glycoproteins prM and E are known to assemble into particles even in the absence of core protein, but their transmembrane (TM) segments form an α -helical hairpin that exposes only a small loop connecting the two TM helices at the cytosolic side. It was not understood, under these conditions, how they recruit the genomic RNA into particles when protein C is present during a normal flavivirus infection. The structure reported here shows that a C protein dimer interacts with a bundle of TM hairpins of a (prM/E)₃ trimer at the cytoplasmic side of the membrane. This (prM/E)₃ trimer appears to be the building block of the assembly – the icosahedral immature particle is formed by 60 such trimers. Because the ectodomains of the three prM/E protomers of this trimer project out of the membrane as an inverted tripod, and each individual tripod leg interacts with one leg of two other adjacent tripods to make a trimeric spike, it has not been clear what the building block was: is it a spike trimer or is it a TM bundle trimer? This paper now shows that the assembly unit is the latter, although both are important for making the intertwined surface lattice of immature flavivirus particles. Another aspect that had not been appreciated is the asymmetry of the interaction, a dimer of core protein contacting asymmetrically the bundle of helical TM hairpins. There are six α -helical hairpins (three prM and three E TM segments), and the C dimer contacts four of them, three from prM and one from E. Furthermore, the C dimers are linked by interactions made by a C-terminal α -helix of C (helix α 5), which corresponds to the signal sequence for translocation of prM into the ER lumen during translation of the viral polyprotein (in which the individual proteins are in the order C-prM-E-NS1-etc.). The C protein has a cleavage site for the viral proteinase NS3 upstream of the α 5 helix, and another important information from this work is that it shows that in virions there is a mixture of cleaved and uncleaved C. Furthermore, the data show that the C proteins forming the lattice of 60 dimers retain α 5, which lies flat on the membrane making antiparallel interactions linking adjacent dimers. The current paradigm posits that signalase cleavage at the C-terminal end of α 5 results in a membrane-anchored C protein (called anchC), which is later released from the membrane by cleavage upstream of α 5 by the viral protease. The negative charge at generated by the new carboxy terminus at the end of α 5 seems to render it unstable as a TM helix (α 5 is also too short to stably span the membrane after cleavage), and thus appears to spontaneously retract from the membrane. The basic stretch that is the recognition site for the viral NS3 protease is probably masked in interaction with phospholipids at the membrane surface. So the present paper provides a clear paradigm shift in our understanding of the whole process, polyprotein maturation and assembly of immature flavivirus particles at the luminal side of the ER membrane.

The authors propose a model for particle assembly in Figure 7, saying that the core assembles first by RNA-bound C dimers interacting with each other via the antiparallel α 5 contacts, which then recruit the (prM/E)₃ bundles to make a particle. This reviewer disagrees with this model, however. Expression of only prM and E leads to formation of empty, closed particles ("virion-like particles" or VLPs), in the absence of C protein. The fact that these VLPs are heterogenous suggest that the core is important for reaching a uniform diameter, but the interactions that drive budding are clearly made by the glycoproteins. And as clearly shown in Figure 6b, the α 5 helices are linked to the dimer core by a flexible linker, indicating that their ordering in the particle is dictated by the interaction of the body of each C dimer with a bundle of TM hairpins of the glycoproteins. In other words, the evidence presented in this paper is much more compatible with a scaffold made by the envelope proteins, each of the 60 bundles recruiting a C protein dimer underneath, in turn bringing in the viral genome. The TM bundles are spaced such that the C-dimers underneath can then interact laterally via the α 5 helices, as observed in the reconstruction. The experiments reported in Fig. 5b with maltose binding protein (MBP) fused to C carrying or not α 5 do not prove point the authors want to make: as α 5 is amphipathic, it is not surprising that they find the ladder of higher aggregates in the gel in the presence and not in the absence of α 5. The information of the gel does not mean that the C proteins are making the same lattice in solution, as the membrane is absent and there is no regularity imposed by the presence of the glycoproteins as in the particle. Most likely, what they see in the gel

corresponds to random aggregates mediated by the hydrophobic and sticky $\alpha 5$ helix.

In summary, the authors make a very important observation that significantly shifts the current paradigm for flavivirus budding and also polyprotein processing, and I find therefore that the results merit publication in Nature Communications. Having said this, there is an issue with their interpretation of the results, and also the way in which the manuscript is written, which will require substantial re-writing to convey a clear message.

Specific issues

- The introduction is too long. The manuscript would benefit of being more concise and going to the point. The real situation is that we know nothing about how the flavivirus genome is recruited into budding particles, and this manuscript provides that information. The introduction should be aimed to presenting this case, not to rehash everything that is known already about flaviviruses in general.
- In the Figures, it would be very helpful to better illustrate how the C dimer is organized. For example, if one of the C subunits in the dimer could be rainbow colored from N- to C-terminus, while the second one is shown in light grey (for instance), this would become clearer (there seem to be some swapping of domains that doesn't stand out, in Figure 2c, right panels, or in Figure 3)
- The way the manuscript is presented is confusing, there is too much space given to the contacts with the antibody, which was useful to maintain the core structured but that by no means constitutes the important information of the paper, which is the organization of the core and the interaction with the bundle of TM segments of the glycoproteins. In this regard, it is unfortunate that the authors were not able to obtain higher resolution. I wonder if they have attempted to make a localized reconstruction centered at the C-prM/E interaction site?

To complete the above sentences, I find that there are way too many Figures and the message of the paper is not clearly stated. Full reconsideration of the manuscript is essential here. But it is worthwhile, as the results are really important.

Reviewer #3:

Remarks to the Author:

In this manuscript, a 9 Å resolution reconstruction of capsid inside the Zika virus has been calculated using the icosahedral symmetry. A carefully validation of the resolution and the map is required for a cryo-EM reconstruction at such resolution. The author used gold standard FSC, asymmetric sub-tomo averaging reconstruction, the comparison between model density and cryo-EM density and the comparison between cryo-EM model and crystal structure to validate and interpret the results. However, the detailed information and the data that the authors provided in the manuscript to validate the map is not enough. The cryo-EM density in the movie does not show many rod-like densities which are normally presented in a cryo-EM map at such resolution to represent the density of alpha helix. It indicates the resolution may be over claimed or some error was induced into the map by averaging or by other unknown reasons.

Major points

1. The author found "capsid proteins do not have full occupancy" in the 16 Å "sub-tomogram asymmetric averaged" map. In addition, as shown in the movie "210136_0_supp_382841_prkybp", the density of the capsid at different site looks different from each other. Putting these together, an asymmetric single-particle reconstruction of Zika-fab complex should be considered.
2. The comparison of density of individual capsid in sub-tomo averaging reconstruction and in single-particle reconstruction is missing. Please provide the FSC curve between two maps of the capsid.
3. The movie "210136_0_supp_382840_prkybk" shows the density of capsid in the 9 Å resolution map. The capsid contains mostly alpha helices which has a rod-like appearance at such resolution. However, I do not see many rod-like densities in the map. The author claimed that "the correlation coefficient between density calculated from the fitted model and the cryoEM map is 0.86". Detail such

as what kind of mask was applied need to be provided. FSC with soft mask instead of a single number should be provided.

4. The position of individual helix in the density map had been refined against the map. However, the RMSD is only 0.94 Å between cryo-EM and crystal models. It is an unusually small number considering that the cryo-EM model had been refined against a 9 Å map. The detail information of the refinement needs to be provided. Such small difference between two structures can be from the error in the map. The small RMSD basically tells that the cryo-EM and crystal structures are very similar. It is strange that the author discussed about the possible reasons which may cause the difference between cryo-EM and crystal structures.

Minor point:

1. The detail information of sub-tomo averaging procedure was not provided in the manuscript. The author claimed a 16 Å "sub-tomogram asymmetric averaged" map. Does the author mean that the capsid part of the virus does not follow the icosahedral symmetry? However, for different virus particles, the overall capsid shell is identical. Therefore, the capsid shell from different virus particles can be averaged without applying any symmetry.

Reviewers' comments:

Reviewer #1 (Remarks to the Author):

The manuscript by Tan et.al., uses multiple techniques to characterize the structure of the Zika virus capsid protein in the immature virion. The structure and orientation of the flavivirus capsid protein within an intact virion has long been elusive. The first indication of some order in the Zika virus capsid organization was presented by Mangala Prasad et.al in 2017. This paper provides a more substantial look into the capsid organization in immature Zika virus by using a Fab complex with the immature virus that stabilizes the capsid layer. The authors also identify that an additional helix of the flavivirus capsid protein is present in the virion which was previously not known. In addition, the authors compare the Fab DV62.5 binding to immature ZIKV and immature DENV to show that the same Fab does not stabilize the imm.DENV capsid. Some comments for the authors:

1. At an overall resolution of 8 Å and a local resolution of 9.2 Å at the capsid protein layer in their Immature ZIKV+Fab map, it is possible to fit structures into the EM map with high confidence but the statement in the abstract - 'All secondary structural elements are visible' is not entirely correct as you cannot separate 'all' secondary structural elements of the capsid protein in their current map without ambiguity.

Corrected in the abstract:

“All secondary structural elements are visible, including a helix...” is now replaced with “Fitting of the capsid protein into densities showed the presence of a helix...”.

2. The authors report a 16 Å resolution sub-tomogram averaged map of the imm.ZIKV+Fab complex to show that some capsid proteins do not have full occupancy in the virus. But there is no description of the method and parameters used in their experimental methods section. This needs to be added and explained.

We added the description of the procedure to the methods section under the header:

“CryoEM tomography data collection, image processing and subtomogram averaging.

A sample of ImmZIKV-Fab DV62.5 was mixed with solution of 10 nm gold particles, put on a thin carbon on lacey carbon coated copper grids, blotted and flash-frozen in liquid ethane.

Tilt series were collected at 47,000x nominal magnification (pixel size 1.71Å), at an underfocus of 3-4 μm. The sample was tilted in sequence from 0° to -30° , then 0° to 60° with 3° step, with a total dose of 90e⁻⁷/Å². Images were collected in movie mode and the frames were combined using MotionCor software³⁴. The tilt series were processed with IMOD³⁵. In total, 63 tilt series were collected and processed. Individual virus particle tomograms were selected with template matching procedure in emClarity³⁶, using a 10x

downsampled 3D map of an icosahedrally averaged immDENV1 cryo-EM structure as a search template. 1934 individual particle tomograms were selected. Subtomogram averaging was carried out in emClarity, with no symmetry applied. After 10 cycles of refinement in emClarity, the resolution for the subtomogram averaged map had reached 16Å. Then, an additional selection of tomograms was done for further refinement. This selection involved first icosahedrally averaging the individual subtomograms with their pre-determined orientation parameters and then compared with the icosahedrally averaged model using Fourier shell correlation (FSC). 1152 particle tomograms showing resolution of better than 40Å at FSC cutoff of 0.5 were selected for further refinement. After another 5 cycles of refinement using these subtomograms with no symmetry applied, the final averaged map was determined to 13Å resolution (reported by emClarity).”

3. Following up with the last comment, the supplemental figure 5 which shows the averaged sub-tomogram map of imm.ZIKV+Fab needs to be annotated more clearly to show that the capsid protein in red ribbons is in an icosahedrally equivalent position to the capsid protein in green ribbons.

We added the following to the legend of Supplementary Fig. 5:

“Capsid proteins related by icosahedral symmetry are shown as ribbons (both red and green).”

4. The panel c in Figure 4 is a bit hard to follow. It would be easier if all the Fab molecules were given a similar color or texture scheme. For example, in the left most panel of Imm.ZIKV+Fab, one set of Fabs are colored in purple and have a smooth shape similar to that of the viral proteins, whereas the Fab which binds near the 3-fold vertex is in grey and has a more rough appearance. It would be useful if the authors could modify the figure so that all the Fab molecules have a distinct rough/bumpy surface so as to separate them clearly from the smoothly shaped viral proteins.

This figure, due to some changes, is now Fig. 2d. We changed the texture to sphere surfaces.

5. The authors state that the DV62.5 Fab binds less strongly to imm. DENV as compared to imm.ZIKV and point to this lower binding as the reason why the imm. DENV capsid is not stabilized by the Fab binding. As the structures of the Fab binding to DENV and ZIKV are very similar, what is the author’s take on why the Fab binds less strongly to imm. DENV compared to imm. ZIKV?

We have observed lower density of the Fab molecule around the 3-fold vertex and hence think that there are partial occupancies on immZIKV, while there is no density for the Fab at this site in the immDENV1. We have done additional work on the immZIKV-Fab DV62.5 complex to figure out the actual occupancies of Fab around the 3-fold vertex by conducting localized reconstruction of this region and found that either one or two Fabs could bind within this region. The new results are added and shown in Fig. 2e.

Regarding the difference in Fab DV62.5 occupancies between immDENV1 and immZIKV; we will reply to Reviewer #1, but no response will be included in the manuscript as we agree with Reviewer #2 and also the editor that we need to keep the content of the manuscript concise. Additionally, since the Fab DV62.5 binding did not improve the dengue capsid density, we decided to remove the entire description of the work on immDENV1:Fab DV62.5 complex.

We determined the affinity of DV62.5 IgG to whole immDENV1 and immZIKV particles by using biolayer interferometry (Octet) and found that their relative affinities are the same. Regardless, we removed the entire section on the immDENV1:Fab DV62.5 complex from the manuscript. The lower Fab occupancy in immDENV1 around the 3-fold vertices is likely due to clashes between the Fab and the neighboring prM-E complex.

See figure below.

- (a) In immZIKV, when Fab (shades of grey) is bound to the light green prM, the Fab does not clash with any other nearby prM-E molecules (see magenta dotted circle).
- (b) In immDENV1 virus, when Fab is bound to the light green prM, it is very close to the neighboring light blue prM molecule, likely resulting in clashing and sub-optimal Fab binding.

6. The unliganded immature ZIKV structure has a partially ordered capsid protein layer whereas the unliganded immature DENV structure shows no such feature. Also, as the authors point out, the ZIKV capsid protein structure is more similar to WNV than DENV. Could a difference in capsid protein sequence and/or structure be responsible for the more stable organization of the capsid protein in immature ZIKV as opposed to DENV? Can the authors comment on it?

Although this is an interesting question, reviewer #2 and the editor suggest that we make the manuscript more concise. Thus, since we did not obtain any capsid structure from immDENV1, we decided to remove all the immDENV1 related parts from the manuscript. We will however, answer Reviewer #1's questions here.

E protein

```
82      90      100     110     120     130     140     150     160
ZIKV  LDKQSDTQYVCKRRTLVDRGWNCGGLFGKGSVTCAKFAKSKKMTGKSTOPENLEYRTMLSVHGSQHSGMIVNDTGHEITDE
WNV   NDKRAIPAFVCRQGVVDRGWNCGGLFGKGSIDTCAKFAKSTKAIGRITLKENIKYEVAIFVHGPTTVESHGNYSTQVGGAT
DENV2 LNEEQDKRRFVCKHSMVDRGWNCGGLFGKGGIVTCAMFTCKNNMKGVVQOPENLEYTIVITPESGEEHAVGNDTGKHGKE

410     420     430     440     450     460     470     480
ZIKV  TIGKAFEAIVRGAKRMAVLGDTAWDFGSGGALNSLGRGIHOIFGAAFKSLFGGMSWFSOILIGTLLMWLGLNNTKNGSISL
WNV   SIGKAFITTLKGAQRLLAALGDTAWDFGSGGVFTSVGKAVHOVFGGAFRSLFGGMSWITQGLIGALLWMLGINARDRSIAL
DENV2 SIGQMIETTMRGAKRMAILGDTAWDFGSLGGVFTSIGKALHOVFGAIYGAAFSGVSWIMKILIGVIIITWIGMNSRSTSLSV
```

prM protein

```
1      10      20      30      40      50      60      70      80
ZIKV  RGSAYYMYLDRNDAGEAISFPTTLGMNKCYSIQIMDLGHMCDATMSYECPMIDEGVEPDDVDCWNTTSTWVYVGTCHHKKG
WNV   FQKVMMTVNATDVTDVITIPATAAGKNLCIVRAMDVGYMCDDTIYECVVISAGNDPBDIDCWNTKSAVYVYGRCTKTR
DENV2 RNGEPHMIVSRQEKGKSLLEFKTEDGVNMCITLMAAMDLECEDETTIKCFELKQNEPBDIDCWNSSTWVYVGTCTTTG

82      90      100     110     120     130     140     150     160
ZIKV  EARRSRRAVTLPSHSTRKIQTRSQTWLESREYTKHLIRVENWIFRNPGFALAAAIWLLGSSTSKVKVIYLVMLLLIAPAY
WNV   HSRRSRRLTVQTHGESTANKKGAWMDSTKATRYLVKTESWILRNPGYALVAAVIGWMLGSNTMORVVFVLLLVAPAY
DENV2 EHRREKRSVALVPHVGMGLERTETWMSSEGAWKHAQRIBETWILRHPCFTIMAAIILAYTIQTHFFQRALIFILLTAVRPSM
```

Capsid protein

```
1      10      20      30      40      50      60      70      80
ZIKV  KKSgGGFRIVNMLKRCVARVSPFFGCH.KRLPAGLLLGHGPIRMVLAhILAFhLRFTAhKPSLGLINRWGSVGGKKEAMEIKKFK
WNV   GPGKSRhAVNMLKRCMPRVLhSLICH.KRAMLSLIDGKGPhIRFVLAhLLAFhFRFTAhIAPTRAVLDRWRGVNKKQTAMKHLhLSFK
DENV2 RKhKARNThPFNMLKREhRNVSTVQQThTKRhFLGLMLQCRhGLKLFMAhLVAFhLRFLhTIPhTAGILKRWGhTIKKSKhINVLhRGR
```

The figure above shows the alignment of flavivirus protein sequences. The interacting interface between E or prM TM region with the capsid protein are boxed in light blue. Red background shows identical residues, red font indicates similar residues, black font indicates non-conserved residues. While residues in the prM and E TM regions are mostly conserved, those corresponding to the capsid protein are more variable.

In the crystal structures of ZIKV (Shang *et al.*, 2018) and WNV (Dokland *et al.*, 2004), although the orientation of helix $\alpha 1$, which is on the interacting interface, is similar, the structure of that helix is different largely because of the presence of two proline residues in ZIKV that disrupt the helical structure. Since all three viruses are different in these areas (either the orientation or structure of helix $\alpha 1$), we cannot draw any conclusions regarding their interactions with E and prM TM regions.

Reviewer #2 (Remarks to the Author):

In this manuscript, Tan et al describe the structure of the Zika virus core particle, determined by cryo-EM to around 9Å resolution. This is the first time the organization of the core of a flavivirus is visualized, showing that it displays icosahedral symmetry and is formed by 60 dimers of the capsid protein (or core protein, termed “C”). The authors used a non-neutralizing antibody against Zika virus that targets an epitope at the prM/E interface distal from the membrane. Other cryo-EM reconstructions of flavivirus immature and mature particles had been reported before, but in all of them,

the core was disordered. The new results are important because they open the way to understand how the viral genome is incorporated into particles during budding. The envelope glycoproteins prM and E are known to assemble into particles even in the absence of core protein, but their transmembrane (TM) segments form an α -helical hairpin that exposes only a small loop connecting the two TM helices at the cytosolic side. It was not understood, under these conditions, how they recruit the genomic RNA into particles when protein C is present during a normal flavivirus infection. The structure reported here shows that a C protein dimer interacts with a bundle of TM hairpins of a (prM/E)₃ trimer at the cytoplasmic side of the membrane. This (prM/E)₃ trimer appears to be the building block of the assembly – the icosahedral immature particle is formed by 60 such trimers. Because the ectodomains of the three prM/E protomers of this trimer project out of the membrane as an inverted tripod, and each individual tripod leg interacts with one leg of two other adjacent tripods to make a trimeric spike, it has not been clear what the building block was: is it a spike trimer or is it a TM bundle trimer? This paper now shows that the assembly unit is the latter, although both are important for making the intertwined surface lattice of immature flavivirus particles.

Another aspect that had not been appreciated is the asymmetry of the interaction, a dimer of core protein contacting asymmetrically the bundle of helical TM hairpins. There are six α -helical hairpins (three prM and three E TM segments), and the C dimer contacts four of them, three from prM and one from E. Furthermore, the C dimers are linked by interactions made by a C-terminal α -helix of C (helix α 5), which corresponds to the signal sequence for translocation of prM into the ER lumen during translation of the viral polyprotein (in which the individual proteins are in the order C-prM-E-NS1-etc.). The C protein has a cleavage site for the viral proteinase NS3 upstream of the α 5 helix, and another important information from this work is that it shows that in virions there is a mixture of cleaved and uncleaved C. Furthermore, the data show that the C proteins forming the lattice of 60 dimers retain α 5, which lies flat on the membrane making antiparallel interactions

linking adjacent dimers. The current paradigm posits that signalase cleavage at the C-terminal end of α 5 results in a membrane-anchored C protein (called anchC), which is later released from the membrane by cleavage upstream of α 5 by the viral protease. The negative charge at generated by the new carboxy terminus at the end of α 5 seems to render it unstable as a TM helix (α 5 is also too short to stably span the membrane after cleavage), and thus appears to spontaneously retract from the membrane. The basic stretch that is the recognition site for the viral NS3 protease is probably masked in interaction with phospholipids at the membrane surface. So the present paper provides a clear paradigm shift in our understanding of the whole process, polyprotein maturation and assembly of immature flavivirus particles at the luminal side of the ER membrane.

We thank the Reviewer #2 for the positive comments on the paper.

The authors propose a model for particle assembly in Figure 7, saying that the core assembles first by RNA-bound C dimers interacting with each other via the antiparallel α 5 contacts, which then recruit the (prM/E)₃ bundles to make a particle. This reviewer disagrees with this model, however. Expression of only prM and E leads to formation of empty, closed particles (“virion-like particles” or VLPs), in the absence of C protein. The fact that these VLPs are heterogenous suggest that the core is important for reaching a

uniform diameter, but the interactions that drive budding are clearly made by the glycoproteins. And as clearly shown in Figure 6b, the $\alpha 5$ helices are linked to the dimer core by a flexible linker, indicating that their ordering in the particle is dictated by the interaction of the body of each C dimer with a bundle of TM hairpins of the glycoproteins. In other words, the evidence presented in this paper is much more compatible with a scaffold made by the envelope proteins, each of the 60 bundles recruiting a C protein dimer underneath, in turn bringing in the viral genome. The TM bundles are spaced such that the C-dimers underneath can then interact laterally via the $\alpha 5$ helices, as observed in the reconstruction.

We thank Reviewer #2 for the insight. We agree with his suggestion and have edited this section to:

“The ability to make virus-like particles (VLPs) by expressing only prM and E proteins (in the absence of capsid proteins) suggests a more active role of these proteins in the virus assembly process²⁶. However, secreted VLPs have irregular sizes and most are smaller (predominantly particles containing only 60 copies of prM-E) than the native virus, suggesting viral assembly involving only E and prM proteins is unable to produce typical flavivirus icosahedral particles containing 180 copies of prM-E. The capsid protein therefore may serve as an important mediator between the surface glycoproteins and viral RNA, ensuring the virus is packaged properly. Here we propose a series of events that may occur during the virus assembly process (Fig. 6): (1) after the virus polyprotein is translated, the capsid proteins on the cytoplasmic side of the ER form homodimers (Fig. 6a) and then organize into a triangular network consisting of trimers of capsid homodimers (Fig. 6b). Simultaneously, on the luminal side of the ER, the prM and E proteins form heterodimers and then three of these heterodimers associate with each other to form a cluster of prM/E complexes resembling an inverted tripod (Fig. 6a). These prM/E inverted tripods then interact with each other (Fig. 6b); (2) On the ER cytoplasmic side, the triangular network of capsid dimers then interact with the TM hairpins of three inverted prM/E tripods forming an assembly unit (Fig. 6b-c). Thus, an assembly unit consists of three capsid dimers interacting with 9 prM/E complexes. (3) The tips of the inverted prM/E tripods from each assembly unit will interact with those from neighboring assembly units at the luminal side of the ER creating a lattice on the ER membrane (Fig. 6d). The functions of the triangular network of capsid proteins could be to (a) provide the correct spacing between the prM-E inverted tripods, (b) bring in the viral genome and (c) bend the membrane to initiate budding of virus particles.”

The experiments reported in Fig. 5b with maltose binding protein (MBP) fused to C carrying or not $\alpha 5$ do not prove point the authors want to make: as $\alpha 5$ is amphipathic, it is not surprising that they find the ladder of higher aggregates in the gel in the presence and not in the absence of $\alpha 5$. The information of the gel does not mean that the C proteins are making the same lattice in solution, as the membrane is absent and there is no regularity imposed by the presence of the glycoproteins as in the particle. Most likely, what they see in the gel corresponds to random aggregates mediated by the hydrophobic and sticky $\alpha 5$ helix.

The higher molecular weight assemblies of the MBP-capsid with helix $\alpha 5$ observed in the western blot correspond to “dimers of dimers” and “trimers of dimers”. These distinct discreet molecular weights do not suggest random aggregation. Nonetheless, we agree with Reviewer #2 that we need to interpret these data with caution, as they are obtained from *in vitro* experiments. We thus added cautionary line at the end of the paragraph that describes the results from the gels:

“One caveat is these are *in vitro* experiments, and thus may not represent the same interactions that occur in the virus particle ie., in the presence of prM/E glycoproteins, lipids and viral RNA.”

In summary, the authors make a very important observation that significantly shifts the current paradigm for flavivirus budding and also polyprotein processing, and I find therefore that the results merit publication in Nature Communications. Having said this, there is an issue with their interpretation of the results, and also the way in which the manuscript is written, which will require substantial re-writing to convey a clear message.

Specific issues

- The introduction is too long. The manuscript would benefit of being more concise and going to the point. The real situation is that we know nothing about how the flavivirus genome is recruited into budding particles, and this manuscript provides that information. The introduction should be aimed to presenting this case, not to rehash everything that is known already about flaviviruses in general.

We agree with Reviewer #2 and have shortened the introduction.

- In the Figures, it would be very helpful to better illustrate how the C dimer is organized. For example, if one of the C subunits in the dimer could be rainbow colored from N- to C-terminus, while the second one is shown in light grey (for instance), this would become clearer (there seem to be some swapping of domains that doesn't stand out, in Figure 2c, right panels, or in Figure 3)

We have corrected Fig. 2c, and Fig. 3c and 3d.

- The way the manuscript is presented is confusing, there is too much space given to the contacts with the antibody, which was useful to maintain the core structured but that by no means constitutes the important information of the paper, which is the organization of the core and the interaction with the bundle of TM segments of the glycoproteins.

We agree with Reviewer #2 and have shortened the results section describing how the Fab stabilizes the immZIKV particle. We also deleted text describing the reconstruction of immDENV1 complexed with Fab DV62.5 since the capsid density did not improve and thus these results provided no additional understanding of capsid protein assembly in dengue.

In this regard, it is unfortunate that the authors were not able to obtain higher resolution. I wonder if they have attempted to make a localized reconstruction centered at the C-prM/E interaction site?

We thank Reviewer #2 for this suggestion, and we have discussed this possibility with Dr. Sjors Scheres (University of Cambridge), the Relion developer. Dr. Scheres cautioned that it would be challenging, if not impossible, given the relatively small mass of the capsid protein relative to the rest of the virus. Nonetheless, we have tried localized refinement as implemented in Relion where we subtracted the prM-E density, and then perform symmetry expansion followed by 3D classification without alignment. We did not obtain a reasonable reconstruction, likely due to: 1) the fact that the capsid dimer is too small in size and 2) it is not possible to correctly subtract the lipid and the RNA density from the experimental images due to the lack a correct model for these regions. We also did focused refinement where we masked the capsid dimer, conducted 3D classification without alignment and subsequently with local refinement. The results we get from these two methods are similarly poor, likely due to the low MW of the capsid protein relative to the virus particle.

To be sure that we implemented the localized refinement procedure correctly, we also applied these methods to the prM-E proteins at the icosahedral 3-fold vertices, and were able to show partial binding of Fabs to some of these glycoproteins around this vertex – see reply to Reviewer #1, question (5) – indicating that the procedure worked properly.

To complete the above sentences, I find that there are way too many Figures and the message of the paper is not clearly stated. Full reconsideration of the manuscript is essential here. But it is worthwhile, as the results are really important.

We have reduced the number of figures (removed one from the main figures and another from the supplementary figure) after re-editing the paper.

Reviewer #3 (Remarks to the Author):

In this manuscript, a 9 Å resolution reconstruction of capsid inside the Zika virus has been calculated using the icosahedral symmetry. A careful validation of the resolution and the map is required for a cryo-EM reconstruction at such resolution. The author used gold standard FSC, asymmetric sub-tomo averaging reconstruction, the comparison between model density and cryo-EM density and the comparison between cryo-EM model and crystal structure to validate and interpret the results. However, the detailed information and the data that the authors provided in the manuscript to validate the map is not enough. The cryo-EM density in the movie does not show many rod-like densities which are normally presented in a cryo-EM map at such resolution to represent the density of alpha helix. It indicates the resolution may be over claimed or some error was induced into the map by averaging or by other unknown reasons.

Major points

1. The author found “capsid proteins do not have full occupancy” in the 16 Å “sub-tomogram asymmetric averaged” map. In addition, as shown in the movie “210136_0_supp_382841_prkybp”, the density of the capsid at different site looks different from each other.

We have now improved the map to 13 Å resolution and have updated the Supplementary Movie 2. The density corresponding to the capsid did not change much compared to the older reconstruction. The individual capsid densities within the asymmetric (C1) sub-tomogram-averaged map looks different from each other, possibly due to the different occupancies of capsid protein in different particle tomograms i.e., some capsid proteins in a specific icosahedrally equivalent positions could be present in all particles, while others may or may not be present. When averaged, the densities of individual capsids within the sub-tomogram averaged map will be different. While the overall resolution of our new C1 subtomogram averaged map is 13Å, the resolution of the capsid layer is 18 Å. Because of the low resolution of the capsid layer, we should not over-interpret the densities. The purpose of the asymmetric sub-tomogram averaged map is to show that the occupancies of capsid proteins at the icosahedrally equivalent positions are not uniform.

Putting these together, an asymmetric single-particle reconstruction of Zika-fab complex should be considered.

We have reconstructed a 12 Å resolution asymmetric single particle analysis (C1 SPA) map of the immZIKV-Fab DV62.5 complex (see below figure); the result is similar to that reconstructed by Therkelsen *et al.* (2018) (~20Å resolution) on the uncomplexed immature Kunjin virus (ImmKUNV) (See below figure). Comparing these C1 SPA maps, our immZIKV-Fab DV62.5 complex map shows clearly the densities of the capsid proteins (colored in orange in below figure) at high contour level, whereas those in the immKUNV quickly disappeared. This suggests that the Fab DV62.5 stabilizes capsid protein positions within the immZIKV. Despite this, our Zika-Fab complex map consistent with the uncomplexed immKUNV maps shows the density of both the E protein layer and the lipid bilayer membrane quickly diminishes on one side of the particle when the contour level is raised.

For the C1 subtomogram averaging of the immZIKV-Fab DV62.5 complex, we have now improved the resolution to 13Å resolution (we have replaced the old figures). The map from our C1 sub-tomogram averaging, in contrast to the C1 SPA maps, shows the E ectodomain layer, the prM-E TM helical and capsid protein densities (local resolution 18 Å) to be evenly strong at different contour levels. In the C1 subtomogram averaging procedure, we first picked subtomograms by template matching using an icosahedral SPA map as a reference. Next, we carried out C1 refinement cycles until the resolution reached 16 Å, and then implemented a selection procedure to choose tomograms with good orientations. This selection procedure involves first imposing icosahedral symmetry to individual subtomograms assuming the determined orientation, and then calculating their correlations to our icosahedrally averaged SPA map. The individual original tomograms (with no symmetry imposed), which show high correlations in this selection procedure, were then used for further C1 refinement cycles. This procedure has been updated in the methods section. It is possible that by including template matching during tomogram picking in the beginning of the reconstruction process, and also the additional selection procedure at mid refinement cycle, we chose tomograms which are highly icosahedral, eliminating those that are architecturally imperfect either because they are slightly broken or as a result of imperfect virus assembly, whereas in C1 SPA procedure, all particles were included.

The C1 SPA immZIKV-Fab DV62.5 complex thus does not add any new information regarding capsid structure beyond what was obtained from our C1 subtomogram-averaged map, and hence we will not include this in our manuscript.

ImmKUNV - [C1 - SPA] (Therkelsen *et. al.*, 2018)

lower contour level

higher contour level

ImmZIKV-Fab DV62.5 complex [C1 - SPA]

lower contour level

higher contour level

ImmZIKV - Fab DV62.5 complex [C1 - Subtomogram Averaging]

lower contour level

higher contour level

2. The comparison of density of individual capsid in sub-tomo averaging reconstruction and in single-particle reconstruction is missing. Please provide the FSC curve between two maps of the capsid.

The C1 sub-tomogram averaged map shows uneven occupancy for the capsid proteins in the different icosahedrally-equivalent positions, thus comparison with the icosahedrally averaged SPA map would not be fair. Additionally, the resolution for the capsid layer in the C1 subtomogram averaged map is 18 Å, hence FSC curve comparison can only be done

meaningfully to a resolution that does not provide meaningful information since it would be comparing between featureless blobs of low resolution densities.

3. The movie “210136_0_supp_382840_prkybk” shows the density of capsid in the 9 Å resolution map. The capsid contains mostly alpha helices which has a rod-like appearance at such resolution. However, I do not see many rod-like densities in the map.

We agree with the reviewer that there is not many rod-like densities, however, in our structure the helices are short except helix $\alpha 4$. This of course results in less rod-like appearance. The longest helix $\alpha 4$ interacts with different parts of the RNA genome and hence may also deviate from the rod like density. We applied a soft mask to the capsid layer for the calculation of resolution by the gold standard FSC using a 0.143 cutoff. It shows a resolution of 9.2 Å and we initially rounded this number to 9 Å; we have now revised the documented and rounded this number up to 9.5 Å .

The author claimed that “the correlation coefficient between density calculated from the fitted model and the cryoEM map is 0.86”. Detail such as what kind of mask was applied need to be provided. FSC with soft mask instead of a single number should be provided.

We used the program Chimera to generate a map from our model to a resolution similar to that of our experimental cryoEM map and then calculate their real space correlation. We did not use FSC for this calculation.

We have added this in the methods section:

“The correlation coefficient between the density calculated from the fitted capsid model and the corresponding density in the cryoEM map showed a value of 0.86. This was determined by using the “Fit in Map” function in Chimera to calculate the correlation coefficient between a simulated 9.5 Å resolution map generated from capsid protein model and the corresponding cryoEM density.”

4. The position of individual helix in the density map had been refined against the map. However, the RMSD is only 0.94 Å between cryo-EM and crystal models. It is an unusually small number considering that the cryo-EM model had been refined against a 9 Å map. The detail information of the refinement needs to be provided.

We have previously described procedure for the fitting and model building of capsid protein into the cryoEM map:

“The capsid protein model was initially fitted into the density as a rigid body dimer and then the fit was optimized by fitting individual helices separately. Further model building and

structure regularization was done in COOT⁴⁰. The prM/E and capsid protein models fitting into uncomplexed immZIKV and immZIKV-Fab DV62.5 complex were refined using the ‘Real-space refinement’ tool⁴¹ in the Phenix programme⁴² with the secondary structure restraint applied resulting in the final overall correlation coefficient of 0.80 and 0.76, respectively.”

Such small difference between two structures can be from the error in the map. The small RMSD basically tells that the cryo-EM and crystal structures are very similar. It is strange that the author discussed about the possible reasons which may cause the difference between cryo-EM and crystal structures.

We thank Reviewer #3 for pointing this out. In the previous manuscript, we used Chimera to superimpose the different capsid protomers of our dimeric structure on to the available NMR and crystal structures and calculated the RMSDs. We examined the results and observed that they use only the best matching residues for RMSD calculations, thus resulting in small RMSD values. Now we have used the CCP4 program (Superpose), which uses secondary structure matching for calculation of RMSD and thus included more residues for comparison. The values have changed from around 1 Å to ~3 Å now (see below and Supplementary Table 1).

PDB ID		Capsid protomers within dimer			
		1		2	
		RMSD (Å)	residues*	RMSD (Å)	residues*
1R6R	DENV2 (NMR structure) chain A	3.31	60	3.31	57
	DENV2 (NMR structure) chain B	3.31	60	3.31	57
1SFK	WNV (crystal structure) chain A	3.33	64	2.55	53
	WNV (crystal structure) chain B	3.33	64	2.53	48
5YGH	ZIKV (crystal structure) chain A	2.89	63	2.35	62
	ZIKV (crystal structure) chain B	2.76	62	2.27	61

*number of aligned residues that were used to calculate RMSD

Minor point:

1. The detail information of sub-tomo averaging procedure was not provided in the manuscript.

We have added into the methods section.

The author claimed a 16 Å “sub-tomogram asymmetric averaged” map. Does the author mean that the capsid part of the virus does not follow the icosahedral symmetry? However, for different virus particles, the overall capsid shell is identical. Therefore, the capsid shell from different virus particles can be averaged without applying any symmetry.

The sub-tomogram asymmetric averaged map was produced by averaging subtomograms without applying any symmetry (C1). We meant the presence of individual capsid proteins may differ between particles (occupancy difference) but when they are present, they are at the icosahedrally equivalent sites.

In flavivirus, the historically named capsid protein is actually more analogous to the RNA binding proteins in other virus systems and does not form a shell, therefore better referred to as capsid layer. The shell is made up of E and prM proteins.

Reviewers' Comments:

Reviewer #1:

Remarks to the Author:

No additional questions or comments for the authors. The manuscript reads better and is more streamlined in the current form.

Reviewer #2:

Remarks to the Author:

The revised manuscript by Tan et al. is much more concise than the previous version. It still has issues though, and below are detailed comments on specific parts of the paper. As I wrote in my review of the previous version, the results described constitute a real landmark in flavivirus research, and there is no question their result should be published. But for the same reason, it is a pity that the authors interpretation goes beyond what their data really shows, which ends up obscuring the message. In particular, they appear to stick with a model in which they propose that trimerization of capsid dimers is required to drive budding, which is not supported by the data. For instance, if this model was true, it would imply that the locations where the capsid protein is missing in particles in which the capsid dimer sites are not fully occupied, would follow a pattern in which those that are occupied would contain always trimers of dimers. Yet their supplementary movie 2, which shows a reconstruction obtained by cryo-ET of the same particle into which the icosahedral symmetry was not imposed, shows that this is clearly not the case: the missing capsid dimers are distributed randomly, not as trimers of dimers. Therefore, it is more straightforward to propose that the dimer interacts with the trimeric bundle of TM hairpins of prM and E, and that as the interactions of the ectodomains of each inverted tripod makes trimeric spikes in the luminal side of the ER, their cytosolic side brings the bound capsid dimers within reach of lateral interactions via their $\alpha 5$ helix, further sealing the assembly. In this way, the assembly will proceed even if not all the TM bundles are occupied by a capsid dimer, resulting in a pattern that is in line with the assembly shown in the supplementary movie 2. This is a beautiful assembly model, and it is a pity that the authors insist in adding steps that are not supported by their own data. The biochemical data obtained in vitro with the MBP fusion, presented in Figures 4 and S9, can have alternative interpretations, especially because of the amphipathic character of helix $\alpha 5$, which will be prone to create the ladders they see in their gel, and cannot be used to support a model that is not supported by their own cET reconstructions. The envelope proteins prM and E are known to be sufficient by themselves to drive budding of closed particles. The presence of the capsid dimers bound to the genome results in homogeneous particles of the same size, which is not the case in the absence of genome. The importance of this manuscript is in showing how the capsid dimers are recruited during budding. There is no need to add arbitrarily additional steps that are not warranted. In conclusion, the paper should be published only if the authors revised their assembly model to match the data presented, and not go beyond it.

In addition, I have listed a number of issues below, which should be addressed:

Lines 123-125: The authors write "The capsid protein densities have tubular shapes corresponding to helical secondary structures present in the crystal and NMR structures of the capsid proteins". I would be more cautious, as in the supplementary movie 1 it is clear that all the alpha-helices of protein C do not fit into tubular densities, and some cut across two tubular densities, going through empty space. This could be attributed to the low resolution, and the authors are right not to try to force each helix into a tubular density. So, all the tubular shapes do not correspond to the observed alpha helices, and stating that the cEM density has tubular shapes that correspond to helical secondary structures of the protein is an overstatement. They could say "some tubular shapes correspond to helices", and that that the structure of a C dimer can be convincingly docked into the cEM density as a rigid body with an

overall good score, resulting in a model that makes biological sense, etc. Also, this paragraph and the following one (lines 148-162) should be condensed into a single one. In addition, it would be important to show what regions of the capsid diverge the most between the X-ray and/or NMR structures and the model obtained after allowing the helices to move independently of each other, a dangerous thing to do at this resolution. It is different to adjust whole domains to fit the EM map (as the authors say it was done with the envelope proteins), than to individually adjust the secondary structure elements within a single domain, as they do here.

Lines 267-268: In principle, it is the envelope proteins that recruit the genomic ribonucleoprotein (RNP) complex to the site of budding, and not the other way around. The authors write here that the capsid dimer recruits prM/E complexes to wherever the genome/C assembly is. It is actually the E/prM complexes that recruit the genomic RNPs to the budding site. This sentence should therefore be reworded.

Lines 251-255: The other way around is also possible, that the dimers bound to the lipid bilayer have the cleavage site for helix $\alpha 5$ protected from cleavage, whereas dimers that only bind RNA do not, and therefore lose $\alpha 5$. There is no need to invoke a requirement for $\alpha 5$ to trimerize to allow dimers to trimerize and interact with the TM regions of surface prM and E proteins, as the authors propose here.

Lines 310–311: Why “some” capsid proteins? Are they not all processed first by the signal peptidase? This would mean that “some” prM would stay uncleaved from the capsid?

And lines 311-312: normally the polyprotein is translated before there is viral replication, so there is no genomic RNA available at the beginning of translation for packaging. Then why propose here that interaction with the RNA is necessary to pull helix $\alpha 5$ out of the membrane? This helix becomes unstable in the membrane and presumably does not need any other interaction, as it is too short and the generation of a negatively charged carboxy-terminus by signalase cleavage within the membrane will be enough to cause its retraction.

Legend to Fig. 2. Please explain the color coding, what is the green, the blue, etc. in the left panels of 2a and b

Line 735 “The center section....” It is not clear what center section the authors mean. It would be better to add an arrow to the Figure, and refer to it to indicate what section they mean.

In Panel 2d, apparently the Fabs bound to the spike to the left of the black triangle were not represented. Maybe this was done for clarity, but there is no explanation in the legend. This should be fixed, otherwise it is confusing.

Line 752: “d” should be “e”. Also, please explain the coloring scheme for E and prM.

Supplementary Figure 9 : How were the gels shown in panels a and b stained? The legend states that the gels in panels c and d were stained with coomassie blue, but there is no information for those in a nor b.

Reviewer #3:

Remarks to the Author:

As I already mentioned in the previous comments, the 9 Å-resolution map lacking the side chain density need to be carefully validated. The capsid protein is composed of several alpha helices, which usually exhibit rod-like densities in a 9 Å-resolution map. By eye inspection, I did not see such features in the map. The map was calculated with icosahedral symmetry applied. Any flexibility in the structure of capsid or the uncertainty of the location of the capsid would cause errors in the averaged

map. Such errors were not completely random and might have caused the over claim of the resolution and the over interpretation of the map. In the worst case, the map can be completely wrong. A conventional technique to validate the map is the tilt pair method. However, in this case, the glycoprotein layer dominates the alignment, therefore is not suitable. The comparison between the fitted crystal structure and the map can give a hint of the quality of the map. However, a simple number as given in the manuscript reported by Chimera program did not really give such information. The results from tomography was less averaged, a carefully comparison between tomography result and single particle result may also help to validate the map. Such data was missing in the manuscript. The differences in occupancy would affect the resolution or the contour of the capsid proteins at different locations in the tomographic reconstruction. It is not enough to explain the capsid proteins in tomographic reconstruction are different from each other. I had point out a risk of over claim the resolution in the previous comments. The author responded by simply decreased the claimed resolution from 9 to 9.5 Å, which is not professional. In addition, since the author used an atomic model to represent the EM map. The assembly mechanism was based on this atomic model. It is very important and common to tell the reader that at which resolution (no matter with what resolution), this map can be represented by the atomic model. Thus, a FSC between the map of capsid and the atomic model of capsid is a key data to validate the atomic model. The FSC data as well as the detailed procedure of calculating the FSC should be provided in the manuscript. Overall, in the revised manuscript, the author failed to improve the validation for both the map and the model, which is necessary for any publication.

Reviewers' comments:

Reviewer #1 (Remarks to the Author):

No additional questions or comments for the authors. The manuscript reads better and is more streamlined in the current form.

Reviewer #2 (Remarks to the Author):

(A) The revised manuscript by Tan et al. is much more concise than the previous version. It still has issues though, and below are detailed comments on specific parts of the paper. As I wrote in my review of the previous version, the results described constitute a real landmark in flavivirus research, and there is no question their result should be published. But for the same reason, it is a pity that the authors interpretation goes beyond what their data really shows, which ends up obscuring the message. In particular, they appear to stick with a model in which they propose that trimerization of capsid dimers is required to drive budding, which is not supported by the data. For instance, if this model was true, it would imply that the locations where the capsid protein is missing in particles in which the capsid dimer sites are not fully occupied, would follow a pattern in which those that are occupied would contain always trimers of dimers. Yet their supplementary movie 2, which shows a reconstruction obtained by cryo-ET of the same particle into which the icosahedral symmetry was not imposed, shows that this is clearly not the case: the missing capsid dimers are distributed randomly, not as trimers of dimers. Therefore, it is more straightforward to propose that the dimer interacts with the trimeric bundle of TM hairpins of prM and E, and that as the interactions of the ectodomains of each inverted tripod makes trimeric spikes in the luminal side of the ER, their cytosolic side brings the bound capsid dimers within reach of lateral interactions via their $\alpha 5$ helix, further sealing the assembly. In this way, the assembly will proceed even if not all the TM bundles are occupied by a capsid dimer, resulting in a pattern that is in line with the assembly shown in the supplementary movie 2. This is a beautiful assembly model, and it is a pity that the authors insist in adding steps that are not supported by their own data.

The biochemical data obtained in vitro with the MBP fusion, presented in Figures 4 and S9, can have alternative interpretations, especially because of the amphipathic character of helix $\alpha 5$, which will be prone to create the ladders they see in their gel, and cannot be used to support a model that is not supported by their own cET reconstructions.

The envelope proteins prM and E are known to be sufficient by themselves to drive budding of closed particles. The presence of the capsid dimers bound to the genome results in homogeneous particles of the same size, which is not the case in the absence of genome. The importance of this manuscript is in showing how the capsid dimers are recruited during budding. There is no need to add arbitrarily additional steps that are not warranted. In conclusion, the paper should be published only if the authors revised their assembly model to match the data presented, and not go beyond it.

We thank reviewer #2 for clarifying his arguments and we agree with his model, which is more in line with our tomography results. We have toned down the importance of capsid protein trimerization in the assembly process.

For the gel, we feel that we should still keep it as reviewer #2 also agree that the amphipathic nature of the helix alpha 5 likely will stimulate interactions between the capsid proteins. Although the experiments are done *in vitro*, in the virus particle once the helix $\alpha 5$ is exposed between neighboring capsid dimers (be it, two or three capsid dimers) they could possibly interact. We have changed the text to below:

~~“Our cryoEM structure shows that uncleaved helix $\alpha 5$ in capsid proteins is important for the organization of three capsid dimers underneath the viral bilayer lipid membrane, and since they interact with the transmembrane regions of the three prM and one E proteins, they likely play an important role in the organization of the virus surface protein shell. To determine ability of helix $\alpha 5$ to induce capsid dimers to form higher oligomers, we purified MBP tagged ZIKV capsid proteins with or without helix $\alpha 5$. In Coomassie stained SDS-PAGE gels (Supplementary Fig. 10c), all samples, whether boiled or not-boiled, showed only the monomeric form. On the other hand, in Western blots using an anti-MBP antibody, we detect some bands of the MBP:capsid protein with helix $\alpha 5$ that correspond to its higher oligomerization states (Fig. 4b) - dimer, dimers of dimers and trimers of dimers. In contrast, the MBP:capsid protein without helix $\alpha 5$, showed largely absence of dimers of dimers and trimers of dimers. We also ran the MPB:capsid protein with and without helix $\alpha 5$ on a native PAGE and the results showed that in the presence of helix $\alpha 5$, a higher oligomer capsid was detected (Fig. 4c). Since the helix $\alpha 5$ is amphipathic in nature, it is not surprising that helix $\alpha 5$ in capsid dimers can interact with each other to form higher oligomers. The asymmetric tomogram averaged map suggests that in some icosahedral equivalent positions, there are missing capsid dimers in the proposed triangular network, but however, if capsid dimers with helix $\alpha 5$ is located close to another capsid dimer, they likely interact via their helix $\alpha 5$ as shown in our icosahedral averaged SPA map.”~~

We have modified the figure and paragraph on the assembly model to:

“The ability to make virus-like particles (VLPs) by expressing only prM and E proteins (in the absence of capsid proteins) suggests a more active role of these proteins in the virus assembly process²⁶. However, secreted VLPs have irregular sizes and most are smaller (predominantly particles containing only 60 copies of prM-E) than the native virus, suggesting viral assembly involving only E and prM proteins is unable to produce typical flavivirus icosahedral particles containing 180 copies of prM-E. The capsid protein therefore may serve as an important mediator between the surface glycoproteins and viral RNA, ensuring the virus is packaged properly. However at the same time as shown in our asymmetric sub-tomogram averaged map (Supplementary Movie 2), capsid dimers occupancies are not full, suggesting that the maintenance of the trimeric interactions of the capsid dimers at all icosahedral equivalent sites is not strictly necessary for assembly. Here we propose a series of events that may occur during the virus assembly process (Fig. 6): (1) after the virus polyprotein is translated, the capsid proteins on the cytoplasmic side of the ER form homodimers (Fig. 6a). Simultaneously, on the luminal side of the ER, the prM and E proteins form heterodimers and then three of these heterodimers associate with each other to form an inverted tripod (Fig. 6a). These prM/E inverted tripods then interact with each other (Fig. 6b); (2) On the ER cytoplasmic side, some of the capsid dimers interact with the TM hairpins of three inverted prM/E tripods. The capsid dimers, which are in close proximity to each other, will make lateral interactions via their helix $\alpha 5$, further strengthening the assembly (Fig. 6c). This forms an assembly unit. (3) The tips of the inverted prM/E tripods from each assembly unit will interact with each other at the luminal side of the ER creating a lattice on the ER membrane (Fig. 6d). The function of capsid protein dimers could be to provide the correct spacing between the prM-E inverted tripods. The prM/E inverted tripods thus attract the capsid protein:RNA complex to the budding site.”

We also changed the Figure 6 to below:

“Fig. 6 | Proposed assembly process of immZIKV particle. a, Capsid proteins are expressed in cytoplasmic side of the ER membrane, where they form homodimers. One side of the homodimer interacts with ER membrane whereas the other side with viral RNA. Simultaneously, on the luminal side of the ER, the prM and E proteins form heterodimers and then three of these heterodimers associate with each other to form an inverted tripod. **b,** The inverted tripods interact with each other. The capsid dimers then attach to the TM regions of the prM/E tripods. **c,** The capsid dimers then interact with other nearby capsid dimers via their helix $\alpha 5$. This sets the spacing between the inverted tripods. Three prM/E inverted tripods with capsid protein dimers binding underneath form an assembly unit. **d,** The tips of these assembly units (pink boxes) then interact to form a virus surface lattice on the lumen side of the ER membrane. One assembly unit is colored in grey and the other in yellow.”

(B) In addition, I have listed a number of issues below, which should be addressed:
 Lines 123-125: The authors write “The capsid protein densities have tubular shapes corresponding to helical secondary structures present in the crystal and NMR structures of the capsid proteins”. I would be more cautious, as in the supplementary movie 1 it is clear that all the alpha-helices of protein C do not fit into tubular densities, and some cut across two

tubular densities, going through empty space. This could be attributed to the low resolution, and the authors are right not to try to force each helix into a tubular density. So, all the tubular shapes do not correspond to the observed alpha helices, and stating that the cEM density has tubular shapes that correspond to helical secondary structures of the protein is an overstatement. They could say “some tubular shapes correspond to helices”,
We have changed accordingly:

“The capsid protein densities have some tubular shapes corresponding to helices present in the crystal and NMR structures of the capsid proteins (Fig. 2c, two left panels, and Supplementary Movie 1).”

and that that the structure of a C dimer can be convincingly docked into the cEM density as a rigid body with an overall good score, resulting in a model that makes biological sense, etc.

We have added these to the manuscript:

“We first fitted the crystal structure of the Zika virus capsid protein ⁷ into the 9.5Å resolution capsid layer density of the Fab DV62.5:immZIKV complexed map obtained from SPA (Fig. 3a) by rigid body fitting. To optimize the fit, individual helices were allowed to move independently to a small degree in order to better fit the experimentally obtained density map; real space refinement was subsequently employed to ensure realistic bond lengths and geometry (Fig. 3a and Supplementary Movie 1). The correlation coefficient between density calculated from the fitted capsid protein model and the corresponding density in the cryoEM map is 0.86 and RMSD of the final fitted model to the ZIKV crystal structure is 2.27 Å (Supplementary Table 1). Structural variation of the cryoEM capsid structure from the crystal structure of recombinant capsid protein is possible, as the helices are connected by flexible loops. Additionally, the crystal structure of the capsid proteins were determined in the absence of lipids and RNA, and therefore may not adopt the same structure as in the virion. For example, it is well known that fitting crystal structures of isolated E proteins into the cryoEM maps of whole virions requires that domains positions are adjusted, likely due to the flexible inter-domain hinge ¹⁸. The final structure makes biological sense, as the hydrophobic helix α 1 is facing the viral lipid membrane and the positively charged helix

α4 towards the RNA genome. The fit largely preserves the overall zika capsid crystal structure (Supplementary Fig. 7a) and has a good correlation score with the density.”

Also, this paragraph and the following one (lines 148-162) should be condensed into a single one.

We fused these two paragraphs together.

(C)In addition, it would be important to show what regions of the capsid diverge the most between the X-ray and/or NMR structures and the model obtained after allowing the helices to move independently of each other, a dangerous thing to do at this resolution. It is different to adjust whole domains to fit the EM map (as the authors say it was done with the envelope proteins), than to individually adjust the secondary structure elements within a single domain, as they do here.

Since the RMSD between the NMR, crystal structures with our cryoEM is not large, it suggests we did not move things around too much. We did not alter the helices, but we did move the helices because they are connected by loops which are highly flexible. When we move the helices, we make sure that the bond length is in the acceptable range.

We have added a new figure in Supplementary Fig. 7a, to show the superposition between the cryoEM and the crystal structure of ZIKV capsid protein- see below.

Supplementary Figure 7 | The ZIKV capsid protein tertiary structure. a, Superposition of the cryoEM ZIKV capsid protein structure with the crystal ZIKV capsid protein structure.

b, Structure comparison of the cryoEM ZIKV capsid protein structure with those of the

crystal structure of ZIKV ^{3,4}, crystal structure of WNV ⁵, and NMR structure of DENV ⁶. The N-terminus (red N letter) and C-terminus (blue C letter) of the capsid protein are indicated.

(D) Lines 267-268: In principle, it is the envelope proteins that recruit the genomic ribonucleoprotein (RNP) complex to the site of budding, and not the other way around. The authors write here that the capsid dimer recruits prM/E complexes to wherever the genome/C assembly is. It is actually the E/prM complexes that recruit the genomic RNPs to the budding site. This sentence should therefore be reworded.

We have added this line to the discussion: “The prM/E inverted tripods thus attract the capsid protein:RNA complex to the budding site.”.

(E) Lines 251-255: The other way around is also possible, that the dimers bound to the lipid bilayer have the cleavage site for helix $\alpha 5$ protected from cleavage, whereas dimers that only bind RNA do not, and therefore lose $\alpha 5$. There is no need to invoke a requirement for $\alpha 5$ to trimerize to allow dimers to trimerize and interact with the TM regions of surface prM and E proteins, as the authors propose here.

We have delete Fig. 4d and any discussion related to this.

(F) Lines 310—311: Why “some” capsid proteins? Are they not all processed first by the signal peptidase? This would mean that “some” prM would stay uncleaved from the capsid? We are sorry for the confusion, all prM would have helix $\alpha 5$ cleaved off eventually. We have now reworded the above lines 310-311 to clarify this:

“It has been shown that there is competition between the signal peptidase and the NS2B/NS3 protease to reach their respective ends of the helix $\alpha 5$, which maybe partially hidden on either side of the ER membrane, thus leading to sequential cleavage ²². Regardless of whether the peptidase or NS2B/NS3 cleave first, it was proposed that the final product is (1) the final mature capsid protein containing residues from the N-terminus to the end of helix $\alpha 4$, and (2) the helix $\alpha 5$ being cleaved off from the N-terminus of the prM molecule.”

Anyway, in the revised version the following has been deleted as suggested by reviewer #2: (1) all discussion about viral RNA interacting with NS2B/NS3 motif on capsid protein preventing NS2B/NS3 cleavage and (2) RNA binding pulling the helix $\alpha 5$ out of the ER membrane (this is done according to reviewer #2 next comment):

~~“We propose that (1) some capsid proteins are first digested by signal peptidase at the C-terminal end of the helix $\alpha 5$, (2) the RNA then interacts with highly positively charged helix~~

~~α 4 and the NS2B/NS3 motif, (3) the formation of this complex then pulls the helix α 5 out of the ER membrane.”~~

(G) And lines 311-312: normally the polyprotein is translated before there is viral replication, so there is no genomic RNA available at the beginning of translation for packaging. Then why propose here that interaction with the RNA is necessary to pull helix α 5 out of the membrane? This helix becomes unstable in the membrane and presumably does not need any other interaction, as it is too short and the generation of a negatively charged carboxy-terminus by signalase cleavage within the membrane will be enough to cause its retraction.

We deleted this discussion. See reply to reviewer #2 (F).

Legend to Fig. 2. Please explain the color coding, what is the green, the blue, etc. in the left panels of 2a and b

The color coding has been added to the figure legend – see below legend.

Line 735 “The center section...” It is not clear what center section the authors mean. It would be better to add an arrow to the Figure, and refer to it to indicate what section they mean.

Maybe the word “center section” is difficult to understand, we now changed it to “quarter of a central slice”. This means a slice of the center of the map.

Fig. 2 | Binding of Fab DV62.5 to immZIKV stabilizes the virus structure resulting in

improved resolution of cryoEM map especially the capsid protein density. a-b, The cryoEM maps of immZIKV uncomplexed and complexed with DV62.5. The surface (left top panels) and a quarter of a central slice (left bottom panels) of the cryoEM maps. The central slice shows the capsid protein bridges the RNA and the inner leaflet of the lipid membrane.

The FSC curves (top right panels) with a 0.143 cutoff showed the overall resolution of the complexed and the uncomplexed immZIKV maps is 8Å and 9Å, respectively. The calculated resolution of the capsid layer region (bottom right panels) showed the capsid densities in the complexed and uncomplexed are determined to 9Å and 11Å, respectively. The map is colored according to its corresponding radius (red: 0-125 Å, gold: 126-157 Å, cyan: 158-201 Å, dark blue: 202-241 Å, and light green : >242 Å). Densities corresponding to Fabs are colored in orange. **c,** Overall structure of a capsid dimer with each protomer containing

helices $\alpha 1$ to $\alpha 5$ (colored from light to dark brown shades). Stereo-diagram (two left panels) of the fitted capsid dimer (ribbon) in its corresponding density (grey). The capsid dimer structure (two right panels) is shown from both side and top views. The N-terminus (red N letter) and C-terminus (blue C letter) of the capsid protein structure are indicated. One of protomer is colored in grey. **d**, Occupancies of the Fab DV62.5 on the immZIKV. Left panel, on immZIKV, Fab DV62.5 (grey surface representation) binds to all red prM-E complex. The epitope on the blue prM-E is completely occluded by neighboring prM-E molecules, therefore no Fab binding is detected. Around the three-fold axis, although the epitope is completely exposed in all green prM-E protein molecules, the space at 3-fold vertices is likely unable to accommodate three Fabs. E-prM proteins with bound Fab DV62.5 at full occupancy are indicated by asterisks (*), whereas those with no binding by crosses (x). The three individual E proteins in an asu, each located adjacent to either 5-, 2- or 3-fold vertices, are colored in red, green and blue, respectively and their binding partner, the prM molecule in a lighter shade of the same color. **e**, Localized reconstruction on the densities surrounding the 3-fold vertices showed two major Fab binding classes. Left panel, the first class of Fab binding shows only one Fab (yellow) around this vertex. Right panel, the second class of Fab binding showed two Fabs bound (yellow and magenta) The E-prM proteins and Fab molecules are drawn in ribbon representation.

In Panel 2d, apparently the Fabs bound to the spike to the left of the black triangle were not represented. Maybe this was done for clarity, but there is no explanation in the legend. This should be fixed, otherwise it is confusing.

The Fig. 2d has been corrected.

Line 752: “d” should be “e”. Also, please explain the coloring scheme for E and prM.

It has been corrected and the explanation on the coloring scheme for E and prM has also been added to the figure legend.

We added the following to Fig. 2d:

“The three individual E proteins in an asu, each located adjacent to either 5-, 2- or 3-fold vertices, are colored in red, green and blue, respectively and their binding partner, the prM molecule in a lighter shade of the same color.”

Supplementary Figure 9 : How were the gels shown in panels a and b stained? The legend states that the gels in panels c and d were stained with coomassie blue, but there is no information for those in a nor b.

Supplementary Fig. 9 is now Supplementary Fig. 10. The gels shown in panels a and b were also stained with coomassie blue. The figure legend has been corrected to include this information:

“Supplementary Figure 10 | Coomassie blue stained SDS–PAGE gel profile of purified immZIKV and the purified recombinant MBP-capsid fusion proteins. a, The immZIKV preparation showed clear bands of E, prM and capsid proteins in a 4 – 20% gradient gel. There is no band that corresponds to M protein (8,415 Da) indicating the virus preparation was highly immature. **b**, In a higher percentage gel (15%), two capsid bands at around 13 kDa were observed. **c**, SDS-PAGE of the recombinant MBP-capsid fusion proteins. Samples were either boiled (+) or not-boiled(-). MBP only, MBP:capsid with helix $\alpha 5$ and MBP:capsid without helix $\alpha 5$ were separated on a 4-20% gradient gel. By Coomassie blue, only one band showing monomeric molecular weight of all samples was detected. Comparison of the intensities of the bands of all samples showed similar amounts had been loaded as in Fig. 4b. **d**, SDS-PAGE of the recombinant MBP-capsid fusion proteins. This is to show that similar amount of proteins had been loaded in the native-PAGE as in Fig. 4c.”

Reviewer #3 (Remarks to the Author):

As I already mentioned in the previous comments, the 9 Å-resolution map lacking the side chain density need to be carefully validated. The capsid protein is composed of several alpha helices, which usually exhibit rod-like densities in a 9 Å-resolution map. By eye inspection, I did not see such features in the map. The map was calculated with icosahedral symmetry applied. Any flexible in the structure of capsid or the uncertainty of the location of the capsid would cause errors in the averaged map. Such errors were not completely random and might have caused the over claim of the resolution and the over interpretation of the map. In the worst case, the map can be completely wrong. A conventional technique to validate the map is the tilt pair method. However, in this case, the glycoprotein layer dominates the alignment, therefore is not suitable. The comparison between the fitted crystal structure and the map can give a hint of the quality of the map. However, a simple number as given in the manuscript reported by Chimera program did not really give such information. The results from tomography was less averaged, a carefully comparison between tomography result and single particle result may also help to validate the map. Such data was missing in the manuscript. The differences in occupancy would affect the resolution or the courter of the

capsid proteins at different locations in the tomographic reconstruction. It is not enough to explain the capsid proteins in tomographic reconstruction are different from each other.

We went back to read the previous comments very carefully, we think now that the reviewer suggested to determine the FSC between the individual capsid densities of the C1 subtomogram averaged map and the SPA icosahedral averaged map. We extracted each of the 60 capsid densities from the C1 sub-tomogram and the icosahedral SPA averaged maps and calculated their FSC. We have added a new figure, Supplementary Fig. 6 (see below).

Supplementary Figure 6 | Fourier shell correlation between the 60 capsid densities from C1 sub-tomogram averaged and the density from the icosahedral SPA averaged maps. Sixty capsid densities were extracted from the C1 sub-tomogram averaged map and then FSC between each of these capsid densities to that of the icosahedral SPA averaged map was calculated. **a**, An example of one of these FSC curves is shown. **b**, Diagram shows plot of resolution at the cutoff at 0.143 of the FSC between each of the individual 60 capsid densities to that in the icosahedral averaged SPA map (purple stars). Most are better than 18 Å

resolution (green line); the resolution determined by gold-standard FSC of the capsid layer of the C1 sub-tomogram averaged map between two-halves tomograms.

I had point out a risk of over claim the resolution in the previous comments. The author responded by simply decreased the claimed resolution from 9 to 9.5 Å, which is not professional.

We did that because it is a standard practice to use the “gold-standard FSC” and this method shows 9.2 Å, we are unable to alter the resolution number. That’s why we rounded the value up - which is still acceptable.

In addition, since the author used an atomic model to represent the EM map. The assembly mechanism was based on this atomic model. It is very important and common to tell the reader that at which resolution (no matter with what resolution), this map can be represented by the atomic model. Thus, a FSC between the map of capsid and the atomic model of capsid is a key data to validate the atomic model. The FSC data as well as the detailed procedure of calculating the FSC should be provided in the manuscript.

We thank reviewer for suggesting this, it can be used to support our resolution claim. We used the program “Phenix-mtriage” to calculate the correlation of our atomic model to the cryoEM map. We extracted the capsid density from the icosahedral averaged SPA map and then Phenix-mtriage generated a model map with our capsid protein coordinates and then conducted FSC. At a cutoff of 0.143, the maps correlated to 8 Å. We included this in Supplementary Fig. 3b.

Phenix also calculates D99 that shows the resolution cutoff beyond which Fourier map coefficients are negligibly small. The determined D99 resolution is also 8 Å.

We have also added this in the methods section:

“The program “Phenix-mtriage”⁴³ was used to calculate the correlation of the capsid protein model to the cryoEM map. We extracted the capsid density from the icosahedral averaged SPA map and then Phenix-mtriage generated a model map from our capsid protein coordinates and then conducted FSC. At a cutoff of 0.143, the maps correlated to 8 Å.”

Overall, in the revised manuscript, the author failed to improve the validation for both the map and the model, which is necessary for any publication.

Reviewers' Comments:

Reviewer #1:

Remarks to the Author:

In Supplemental Fig. 3b, when calculating the correlation between an experimental map and its fitted model, a threshold of FSC 0.5 is normally acceptable. An FSC cutoff of 0.143 is applicable only when the two data being compared are completely independent, which is not the case for a fitted model and its map. At the least, it would be good if the authors showed the FSC cutoff values at 0.5 and 0.143.

Phenix mtriage gives out a few different statistics like d99, d_model and d_FSC_model. The authors only mention d99 in their response to reviewer, which indicates the resolution cutoff beyond which the map coefficients are negligible, but d_model (resolution cutoff at which the model map is the most similar to the experimental map) and d_FSC_model (resolution cutoff up to which the model and map Fourier coefficients are similar) are the more appropriate statistics.

Secondly, in supplemental fig. 3, the authors have panel A as FSC curve between 2 half-maps of the prm-E layer of the imm.ZIKV-Fab map. And panel b, as the model-to-map FSC curve of the capsid layer and capsid model. The more appropriate comparison would be to have the FSC curve of the 2 half-maps of the capsid layer put through Phenix mtriage as panel A and then the capsid layer to model FSC curve from phenix mtriage. This way, the statistical values output by Phenix mtriage would be directly assessable and comparable.

From the perspective of the whole manuscript, the fit of the capsid model to the map is important for the further assembly model of the capsid proteins that the authors have proposed. The crux of the capsid protein arrangement in the manuscript is based on the position of helix alpha-5, which the author have observed and built into the EM map. A point to note is that even if the resolution of the EM map in the capsid layer is close to 9A, it is different from the FSC cutoff value in Suppl Fig.3b which only estimates the fit of model to map. Thus, the building or placing of the alpha-5 helix in the imm.ZIKV map is okay and so is the assembly model hypothesized; but it still doesn't imply that the whole of the capsid protein density is interpretable in the experimental map. A modification of the manuscript showing the cutoff at FSC 0.5 for the Phenix mtriage results in Suppl. Fig.3b and applicable modifications of the derived results is needed.

Reviewer #2:

Remarks to the Author:

The authors have revised the manuscript to avoid overinterpretation of their data, which was may main concern with the previous version. The model for assembly they propose now is much more consistent with the data provided.

Reviewer #3:

Remarks to the Author:

Since the model is built according to the density of the map, threshold of 0.5 is normally used for the FSC between a model and a map. Current data shows that the model poorly represent the map.

Reviewers' comments:

Reviewer #1 (Remarks to the Author):

In Supplemental Fig. 3b, when calculating the correlation between an experimental map and its fitted model, a threshold of FSC 0.5 is normally acceptable. An FSC cutoff of 0.143 is applicable only when the two data being compared are completely independent, which is not the case for a fitted model and its map. At the least, it would be good if the authors showed the FSC cutoff values at 0.5 and 0.143.

In the last version of the “reply_to_reviewers”, we calculated the FSC between the map and the fitted capsid model by using phenix. The fitted model that was used included side chains, in this version to be conservative, we removed the side chains of the fitted model as at 9Å resolution, it is not possible to fit side chains. We also included a control -- calculation of the correlation between the C α chain of the fitted pr-E protein molecules with its corresponding EM density [although this density is higher in resolution (6.5Å) than the capsid layer (9.5Å)], where the fit of pr-E density is clearly good by visual inspection – see below.

Fit of pr-E into its corresponding pr-E layer density (6.5Å)

The new FSC curves for both pr-E and capsid protein fitted models to their respective densities is updated in the Supplementary Fig. 3 (panel c and d) – see below. At FSC cutoff of 0.5, they correlate to 9Å and ~10.7Å, respectively, both calculated by using the same program (Phenix Mtriage).

Supplementary Figure 3 | Resolution of the pr-E and capsid layers of immZIKV-Fab DV62.5 map calculated by the gold standard FSC curves, and the FSC between the fitted structure models and their corresponding densities. a-b, The gold standard FSC curve with a 0.143 cut-off between two halves sets of the data showed the resolution of the pr-E layer is 6.5 Å (calculated by Phenix mtriage), whereas the capsid layer is 9.2 Å (both calculation using Phenix Mtriage³ and EMAN (Fig. 2a) showed the same resolution). **c-d,** The FSC curves showed the (c) fitted pr-E proteins and (d) capsid models to their corresponding densities at FSC cut-off of 0.5 correlated to ~9.1 and ~10.7 Å, respectively. All FSC curves were calculated using Phenix.mtriage and the statistics of the mtriage results are shown in Supplementary Table 2.

We have updated the Results, section “**Fab DV62.5 binding improved capsid protein densities in virus**”, to:

“HMAb DV62.5 was previously shown to cross-react to all four serotypes of DENV and ZIKV and is non-neutralizing¹⁷. Here we determined the icosahedrally averaged cryoEM structure of an immZIKV complexed with a Fab DV62.5 (Fig. 2a, left panels) by single particle analysis (SPA). The overall resolution of the map is 8 Å, as determined by the gold standard Fourier shell correlation method (Fig. 2a, top right

panel). The resolutions of prM-E layer and the capsid protein layer of the map are $\sim 6.5\text{\AA}$ (Supplementary Fig. 3a) and $\sim 9.5\text{\AA}$ resolution (Fig. 2a, bottom right panel, and Supplementary Fig. 3b), respectively. The capsid protein densities have some tubular shapes corresponding to helices present in the crystal and NMR structures of the capsid proteins (Fig. 2c, two left panels, and Supplementary Movie 1). We also determined the structure of the uncomplexed immZIKV to overall 9\AA resolution (Fig. 2b). Consistent with the uncomplexed immZIKV structure previously solved by Prasad and colleagues⁸, the capsid protein density for this structure is poor, indicating either inherent flexibility of capsid proteins and/or different relative positions in individual particles (Fig. 2b, bottom left panel, and Supplementary Fig. 4, bottom panels). These results indicate that Fab DV62.5 stabilizes the immZIKV capsid proteins, resulting in well-resolved capsid density to allow fitting with the crystal structure of capsid protein (Fig. 3a and Supplementary Movie 1). However, the density of the capsid protein layer compared to that of the prM-E is slightly weaker, as the capsid layer disappeared at lower visualization thresholds than the prM-E layer. We collected tomograms of immZIKV:Fab DV62.5 complex and calculates an $\sim 16\text{\AA}$ resolution sub-tomogram asymmetric averaged map that shows that the capsid proteins do not have full occupancy at some of the icosahedral symmetry equivalent positions in the capsid protein layer (Supplementary Fig. 5 and Supplementary Movie 2). FSC calculation between each of the 60 capsid densities in the C1 subtomogram averaged map and the density from the icosahedral averaged SPA cryoEM map showed that most of the capsid densities from C1 subtomogram averaged map correlate to the density from the SPA cryoEM map to resolution better than 18\AA , with some capsid densities even reached $\sim 10\text{\AA}$ (Supplementary Fig. 6a,b). There are also capsid densities with correlation $>20\text{\AA}$. These low correlations are from the capsid protein dimer with partial occupancy or even missing leading to poor densities. In the SPA cryoEM map, this partial occupancy likely explains the weaker density of the capsid protein compared to the prM-E layer. We first fitted the crystal structure of the Zika virus capsid protein⁷ into the 9.5\AA resolution capsid layer density of the Fab DV62.5:immZIKV complexed map obtained from SPA (Fig. 3a) by rigid body fitting. To optimize the fit, individual helices were allowed to move independently to a small degree in order to better fit the experimentally obtained density map; real space refinement was subsequently employed to ensure realistic bond lengths and geometry (Fig. 3a and Supplementary Movie 1). The correlation coefficient between density

calculated from the fitted capsid protein model and the corresponding density in the cryoEM map is 0.86 and RMSD of the final fitted model to the ZIKV crystal structure is 2.27 Å (Supplementary Table 1 and Supplementary Fig. 7a). **FSC calculation between the capsid density map and the map generated from its fitted C α chain model (see Methods for details) showed that, at cutoff 0.5, the maps are correlated to 10.7 Å (Supplementary Table 2 and Supplementary Fig. 3d).** Structural variation of the cryoEM capsid structure from the crystal structure of recombinant capsid protein is possible, as the helices are connected by flexible loops. Additionally, the crystal structure of the capsid proteins were determined in the absence of lipids and RNA, and therefore may not adopt the same structure as in the virion. For example, it is well known that fitting crystal structures of isolated E proteins into the cryoEM maps of whole virions requires that domains positions are adjusted, likely due to the flexible inter-domain hinge¹⁸. The final structure makes biological sense, as the hydrophobic helix α 1 is facing the viral lipid membrane and the positively charged helix α 4 towards the RNA genome. The fit largely preserves the overall zika capsid crystal structure (Supplementary Fig. 7a) and has a good correlation score with the density.”

We have updated the Methods, section: **Fitting of prM-E and capsid structure into cryoEM density maps**, to:

“The model building in the cryoEM maps of uncomplexed immZIKV and immZIKV-Fab DV62.5 complex were done by fitting the crystal structure of ZIKV capsid protein (PDB ID: 5YGH) and homology models of E protein, prM, and heavy and light chains of Fab DV62.5 that were prepared separately using Swiss-model server³⁹. The immature dengue virus serotype 1 structure (PDB ID: 4B03) was used as template to build the homology model of immZIKV E and prM proteins, whereas two human antibody structures (PDB IDs: 5AZE and 4JZN) were used as template for the heavy and light chains of Fab DV62.5, respectively. The fitting of E and prM proteins into the cryoEM maps of immZIKV-Fab DV62.5 complex and uncomplexed immZIKV were done using the ‘Fit in Map’ tool in Chimera programme⁴⁰. The Fab DV62.5 model was also fitted into the density in the cryoEM map of immZIKV-Fab DV62.5 complex using Chimera. There are two possible orientations of the Fab molecules in the density, but one of the two fits resulted in a higher correlation coefficient (0.92 vs 0.87). Surface charge analysis of the interacting surfaces in the

final fitted structure showed charge complementarity between the Fab paratope and the epitope (Supplementary Fig. 8c). The surface charge analysis was done in Chimera. The model building of capsid protein into the immZIKV-Fab DV62.5 complex maps calculated with different level of sharpening (B factors of -900 and -550 Å²) was done manually using the programme O⁴¹. The capsid protein model was initially fitted into the density as a rigid body dimer and then the fit was optimized by fitting individual helices separately. Further model building and structure regularization was done in COOT⁴². The prM/E and capsid protein models fitting into uncomplexed immZIKV and immZIKV-Fab DV62.5 complex were refined using the ‘Real-space refinement’ tool⁴³ in the Phenix programme⁴⁴ with the secondary structure restraint applied resulting in the final overall correlation coefficient of 0.80 and 0.76, respectively. The correlation coefficient between the density calculated from the fitted capsid model and the corresponding density in the cryoEM map showed a value of 0.86. This was determined by using the “Fit in Map” function in Chimera to calculate the correlation coefficient between a simulated 9.5 Å resolution map generated from capsid protein model and the corresponding cryoEM density. The program “Phenix.mtriage”³⁴ was used to calculate the correlation of the fitted protein models to the cryoEM map. We extracted the densities of capsid protein and pr-E proteins from the icosahedrally averaged SPA map and then Phenix-mtriage generated the model maps from C α backbone of fitted molecules and conducted FSC calculation. At a cutoff of 0.5, the models correlated to 10.7 and 9.1 Å, respectively (Supplementary Table 2 and Supplementary Fig. 3c and d). Structural alignment of our cryoEM ZIKV capsid structure to the NMR DENV, crystal WNV, and crystal ZIKV structures to calculate the RMSD values was done by using Superpose program⁴⁵.

Phenix mtriage gives out a few different statistics like d99, d_model and d_FSC_model. The authors only mention d99 in their response to reviewer, which indicates the resolution cutoff beyond which the map coefficients are negligible, but d_model (resolution cutoff at which the model map is the most similar to the experimental map) and d_FSC_model (resolution cutoff up to which the model and map Fourier coefficients are similar) are the more appropriate statistics.

A new Supplementary Table 2 showing the Phenix.mtriage summary has been added. The table (see below) contains the statistics including d99, d_model and d_FSC_model.

Supplementary Table 2 | Phenix Mtriage summary

Map resolution estimates	pr-E	Capsid
using map alone (d99) [#]	8.90	8.08
Overall B _{iso}	470.00	495.00
comparing with model (d _{model}) [§]	8.20	17.00
comparing with model (d _{model_b0}) b _{iso_overall} =0	9.60	10.00
d _{fsc_model} *		
FSC(map,model map)=0	6.97	5.82
FSC(map,model map)=0.143	7.79	7.63
FSC(map,model map)=0.5	9.06	10.65

[#]d99 - resolution cutoff beyond which Fourier map coefficients are negligibly small. Calculated from the map.

[§]d_{model} - resolution cutoff at which the model map is the most similar to the target (experimental) map.

*d_{FSC_model} - resolution cutoff up to which the model and map Fourier coefficients are similar.

Secondly, in supplemental fig. 3, the authors have panel A as FSC curve between 2 half-maps of the pr-E layer of the imm.ZIKV-Fab map. And panel b, as the model-to-map FSC curve of the capsid layer and capsid model. The more appropriate comparison would be to have the FSC curve of the 2 half-maps of the capsid layer put through Phenix mtriage as panel A and then the capsid layer to model FSC curve from phenix mtriage. This way, the statistical values output by Phenix mtriage would be directly assessable and comparable.

We thank reviewer for suggesting this. In addition, we thought we should also include a positive control -- calculating the pr-E layer model to its map density for comparison. All calculations for the new Supplementary Fig.3 are done using Phenix.mtriage, in (a) and (b) we calculated the gold standard FSC between two half maps of pr-E and capsid layer, respectively and at FSC cutoff of 0.143, their resolutions are 6.5 Å and 9.2 Å, respectively. (c) and (d) are the FSC curve correlation between pr-E and capsid model maps to their respectively cryoEM density maps and at 0.5 FSC cutoff, they correlated to 9.1 Å and 10.7 Å, respectively.

We have updated the Methods, section “CryoEM data collection, image processing and 3D reconstruction”, to:

“Vitrified viruses were imaged using a 300kV Titan Krios microscope (FEI) fitted with a Falcon II direct electron detector (FEI). Automated image data collection was carried out using Legion software^{28,29}. For uncomplexed immZIKV, 3366 micrographs at 59,000x magnification with pixel size of 1.34 Å, were collected. For

immZIKV-Fab DV62.5 complex, 2,323 micrographs at 47,000x magnification with pixel size of 1.71 Å were collected. Contrast transfer function parameters were estimated using the GCTF programme³⁰. Micrographs showing significant astigmatism and drift were discarded. Only micrographs within the defocus range of 0.5 to 3.5 µm were included for further processing. For each image dataset, initial particle picking was done manually for approximately 1000 particles using the e2boxer function in EMAN2³¹ and then 2D classification was done on these particles using Relion2.1³². Representative 2D class averages derived from manually picked particles were used as reference for automated particle picking using the Gautomatch software (downloaded from <http://www.mrc-lmb.cam.ac.uk/kzhang/>). Further cryoEM reconstruction processing was done using Relion2.1. Two rounds of 2D classification were carried out to remove broken particles. Three dimension classification with particle alignment and icosahedral symmetry imposed was performed using the cryoEM map of immZIKV (EMD-8508) as starting model. Particles in good 3D class averages were combined and final 3D reconstruction was done. The final cryoEM maps were subjected to B-factor sharpening in Relion2.1. In total, 7922 particles were selected in the reconstruction of the uncomplexed immZIKV, while 19295 particles were selected to reconstruct the immZIKV-Fab DV62.5 complex. The resolution of each map was estimated using 0.143 cutoff of the gold standard Fourier shell correlation curve calculated from two maps that were reconstructed from independent two-half datasets. The all FSC calculation in Figure 2 was done using EMAN³³. In Supplementary Fig. 3, all FSC calculations are repeated using Phenix.mtriage – the resolution determined for these maps are similar between programs. For resolution estimation for the capsid and the E ectodomain layers in the uncomplexed immZIKV and the immZIKV:DV62.5 Fab complex maps, we applied a soft-edge radial shell mask either covering capsid or E ectodomain-pr densities layer.”

From the perspective of the whole manuscript, the fit of the capsid model to the map is important for the further assembly model of the capsid proteins that the authors have proposed. The crux of the capsid protein arrangement in the manuscript is based on the position of helix alpha-5, which the author have observed and built into the EM map. A point to note is that even if the resolution of the EM map in the capsid layer is close to 9Å, it is different from the FSC cutoff value in Suppl Fig.3b which only estimates the fit of model to map. Thus, the building or placing of the alpha-5 helix in the imm.ZIKV map is okay and so is the assembly model hypothesized; but it still doesn't imply that the whole of the capsid protein density is interpretable in the experimental map. A modification of the manuscript showing the cutoff at FSC 0.5

for the Phenix mtriage results in Suppl. Fig.3b and applicable modifications of the derived results is needed.

We thank reviewer #1 for the positive comments and clarifications. This is a very important point; the resolution required depends on the question being asked. The model we are presenting does not depend on high resolution information and thus the essential features of the model as presented would not change significantly if the map were much higher resolution.

All requested experiments and changes to manuscript have been done – see all answers above.

Reviewer #2 (Remarks to the Author):

The authors have revised the manuscript to avoid overinterpretation of their data, which was may main concern with the previous version. The model for assembly they propose now is much more consistent with the data provided.

Reviewer #3 (Remarks to the Author):

Since the model is built according to the density of the map, threshold of 0.5 is normally used for the FSC between a model and a map. Current data shows that the model poorly represent the map.

Please see all replies to reviewer #1. Hope that clarifies.

Reviewers' Comments:

Reviewer #1:

Remarks to the Author:

The authors have revised the manuscript and included additional details regarding their fitting and model building. I have no further comments regarding the manuscript.

Reviewer #3:

Remarks to the Author:

The model VS map FSC looks fine in this revised manuscript and is much better than the one in the previous revised manuscript. Authors mentioned in the paper that they deleted the side chains in the model before calculating the FSC, which is perfect fine. However, one thing that was not described in the manuscript is that what kind of mask the authors applied to the map before calculating the FSC and how they created this mask. Because a tight mask calculated based on the atomic model would induce extra model bias, which would significantly improves the model-VS-map FSC.

Reviewers' comments:

Reviewer #3

The model VS map FSC looks fine in this revised manuscript and is much better than the one in the previous revised manuscript. Authors mentioned in the paper that they deleted the side chains in the model before calculating the FSC, which is perfect fine. However, one thing that was not described in the manuscript is that what kind of mask the authors applied to the map before calculating the FSC and how they created this mask. Because a tight mask calculated based on the atomic model would induce extra model bias, which would significantly improves the model-VS-map FSC.

A soft mask (see figure below in blue mesh) was calculated from the $C\alpha$ chain model by Phenix.mtriage with default settings. The mask covers the model sufficiently. Thus, the mask should not introduce much bias to the model-VS-map FSC calculation.

We have added the information regarding the mask that we used for the calculation FSC between map and model:

Section: **Fitting of prM-E and capsid structure into cryoEM density maps**

The program “Phenix.mtriage”⁴⁴ was used to calculate the correlation of the fitted protein models to the cryoEM map. We extracted the densities of capsid protein and pr-E proteins from the icosahedrally averaged SPA map and then Phenix-mtriage generated the model maps from $C\alpha$ backbone of fitted molecules and conducted FSC calculation. A soft mask was calculated using default settings in Phenix.mtriage⁴⁴. At a cutoff of 0.5, the models correlated to 10.7 and 9.1 Å, respectively (Supplementary Tabel 2 and Supplementary Fig. 3c,d). Structural alignment of our cryoEM ZIKV capsid structure to the NMR DENV, crystal WNV, and crystal ZIKV structures to calculate the RMSD values was done by using Superpose program⁴⁵.